# High-resolution mapping of urban air quality with heterogeneous observations: a new methodology and its application to Amsterdam

Bas Mijling[1]

[1]Royal Netherlands Meteorological Institute (KNMI), Postbus 201, 3730 AE, De Bilt, The Netherlands

*Correspondence to*: Bas Mijling (mijling@knmi.nl)

**Abstract.** In many cities around the world people are exposed to elevated levels of air pollution. Often local air quality is not well known due to the sparseness of official monitoring networks, or unrealistic assumptions being made in urban air quality models. Low-cost sensor technology, which has become available in recent years, has the potential to provide complementary information. Unfortunately, an integrated interpretation of urban air pollution based on different sources is not straightforward

because of the localized nature of air pollution, and the large uncertainties associated with measurements of low-cost sensors. This study presents a practical approach to producing high spatio-temporal resolution maps of urban air pollution capable of assimilating air quality data from heterogeneous data streams. It offers a two-step solution: (1) building a versatile air quality model, driven by an open source atmospheric dispersion model and emission proxies from open data sources, and (2) a practical spatial interpolation scheme, capable of assimilating observations with different accuracies.

The methodology, called Retina, has been applied and evaluated for nitrogen dioxide ($NO_2$) in Amsterdam, the Netherlands, during the summer of 2016. The assimilation of reference measurements results in hourly maps with a typical accuracy (defined as the ratio between the root means square error and the mean of the observations) of 39% within 2 km of an observation location, and 53% at larger distances. When low-cost measurements of the Urban AirQ campaign are included, the maps reveal more detailed concentration patterns in areas which are undersampled by the official network. It is shown that during the

summer holiday period, $NO_2$ concentrations drop about 10%. The reduction is less in the historic city centre, while strongest reductions are found around the access ways to the tunnel connecting the northern and the southern part of the city, which was closed for maintenance. The changing concentration patterns indicate how traffic flow is redirected to other main roads.

Overall, it is shown that Retina can be applied for an enhanced understanding of reference measurements, and as a framework to integrate low-cost measurements next to reference measurements in order to get better localized information in urban areas.

**1 Introduction**

Due to growing urbanization in the last decades, more than half of the world's population lives in cities nowadays. Dense traffic and other human activity, in combination with unfavourable meteorological conditions, often cause unhealthy air pollution concentrations. Over 80% of the urban dwellers are forced to breathe air which does not meet the standards of the World Health Organization (WHO, 2016). In 2015, an estimated 4.5 million people died prematurely from diseases attributed

to ambient air pollution (Lelieveld et al., 2018). Good monitoring is important to better understand the local dynamics of air pollution, to identify hot spots, and to improve the ability to anticipate events. This is especially relevant for nitrogen dioxide ($NO_2$) concentrations, which can vary considerably from street to street. $NO_2$ is, apart from being a toxic gas on its own, an important precursor of particulate matter, ozone, and other regional air pollutants. Observations from a single location are not necessarily representative for a larger area. Unfortunately, urban air quality reference networks are usually sparse or even absent due to their high installation and maintenance costs. New low-cost sensor technology, available for several years now, has the potential to extend an official monitoring network significantly, even though the current generation of sensors have significant lower accuracy (WMO, 2018).

However, exploiting these measurements (either official or unofficial), apart from publishing the data as dots on a map, is not straightforward. Here, the aim is to make better use of the existing measurement networks to get the best description of hourly urban air quality, and to create value from low-cost measurements towards a Level 4 product, according to the classification proposed by Schneider et al. (2019)

To obtain high-resolution information of air pollutants with sharp concentration gradients, a very sparse observation network needs to be accompanied by a valid high-resolution air quality model, whereas a very dense network can do with simple spatial interpolations. The situation in most large cities is somewhere in between. There is often a reasonably large reference network present (10+ stations), sometimes complemented with an experimental network of low-cost AQ sensors. Assumptions about underlying unresolved structures in the concentration field are still needed, but this can be done with a simplified air quality model, using the available measurements to correct simulation biases where needed.

A popular approach in detailed mapping of air quality is land use regression modelling (LURM), see e.g. Beelen et al. (2013). LURM uses multiple linear regression to couple a broad variety of predictor variables (geospatial information such as traffic, population, altitude, land use classes) to the observed concentrations. It is typically used in exposure studies, which target long integration intervals by definition. Problems of over-fitting might arise when too many predictor variables are used. Alternatively, Denby (2015) advocates the use of less proxy data, and a model based on more physical principles. In his approach, the emission proxies are first (quasi) dispersed with a parameterized inverse distance function, before being coupled to observed concentrations in a regression analysis.

Mapping of air pollution for short time scales is challenging. Only a few scientific studies are published aiming at assimilation of near-real time observations in hourly urban concentration maps. Tilloy et al. (2013) use the 3-hourly output of a well-developed implementation of the AMDS Urban dispersion model in Clermont-Ferrand, France, to assimilate in-situ $NO_2$ measurements at 9 reference sites in an optimal interpolation scheme. With a leave-one-out validation they show a strong reduction in root mean square error of the time series after assimilation. Schneider et al. (2017) use Universal Kriging to combine hourly $NO_2$ observations of 24 low-cost sensors in Oslo, Norway, with a time-invariant basemap. The basemap is created from a yearly average concentration field calculated with an Eulerian/Lagrangian dispersion model on a 1 km grid, downscaled to 100 m resolution. Averaged over reference locations, their study shows that hourly values compare well with official values, showing the potential of low-cost sensor data for complementary air quality information at these time scales.

In this paper presents a more advanced yet practical approach to map hourly air pollutant concentrations, named Retina. Its main system design considerations are:

- Observation driven
- Able to assimilate observations of different accuracy
- Potential near-real time application
- Versatile / portable to other domains
- Based on open data
- Reasonable computer power

The method is applied to Amsterdam, a city like many where $NO_2$ emissions are dominated by transport and residential emissions and where local exceedances of limit values are regularly observed. Amsterdam is the most populous city in the Netherlands, with an estimated population of 863,000. Located at 52°22′N 4°54′E, it has a maritime climate with cool summers and moderate winters. Concentrations of $NO_2$ within the city vary considerably, being partly produced locally and partly transported from outside the city. Measurements of 2016 show that, compared with regional background values from the CAMS ensemble (see Section 3.2.3), urban background concentrations are on average around 50% higher, while at road sides $NO_2$ concentrations are about 100% higher.

Retina uses a two-stage approach. It runs an urban air quality model to account for hourly variability in meteorological conditions (described in Section 3) which is dynamically calibrated with recent measurements (Section 4). In the second stage it assimilates current measurements using statistical interpolation (Section 5). Section 6 presents the validation of the system, while Section 7 shows the added value when assimilating additional low-cost sensor measurements. The last section is reserved for discussion, conclusion and outlook.

## 2 Air quality measurements

The Public Health Service of Amsterdam (GGD) is the responsible authority for air quality measurements in the Amsterdam area. Within the domain used in this study their $NO_2$ network consists of 15 reference stations: 5 stations classify as road station, 5 as urban background station, 2 as industry, 2 as rural, and 1 undecided. Alternatingly, GGD operates a Teledyne API 200E and a Thermo Electron 42I $NO/NO_X$ analyser, both based on chemiluminescence. A catalytic-reactive converter converts $NO_2$ in the sample gas to NO, which, along with the NO present in the sample is reported as $NO_X$. $NO_2$ is calculated as the difference between $NO_X$ and NO. The accuracy of both type of reference instruments is estimated at 3.7% (GGD, 2014), following the EN 14211 standard which includes all aspects of the measurements method: uncertainties in calibration gas and zero gas, interfering gases, repeatability of the measurement, derivation of $NO_2$ from $NO_X$ and NO, and averaging effects.

Low-cost $NO_2$ measurements are taken from the 2016 Urban AirQ campaign (Mijling et al., 2018). Sixteen low-cost air quality sensor devices were built and distributed among volunteers living close to roads with high traffic volume for a 2-month measurement period, from 13 June to 16 August. The devices are built around the NO2-B43F electrochemical cell by Alphasense Ltd (Alphasense, 2018). The sensor generates an electrical current when the target gas diffuses through a membrane where it is chemically reduced at the working electrode. Better sensor performance at low ppb levels is obtained by

using low-noise interface electronics. The sensor devices were carefully calibrated using side-by-side measurements next to a reference station, solving issues related to sensor drift and temperature dependence (Mijling et al., 2018). After calibration,

they are found to have a typical accuracy of 30%.

## 3 Setting up a versatile urban air quality model

One of the largest unknowns when modelling urban air quality is a detailed, up-to-date emission inventory capable of describing the local contribution. For cities such as Amsterdam the local emissions are dominated by the transport and residential sector. This is confirmed by the EDGAR HTAP v2 emission inventory (Janssens-Maenhout et al., 2013), which

estimates the contribution of $NO_X$ emissions in a $20 \times 33$ km$^2$ (0.3 degree) area around the centre being 62%, 20%, 12%, and 6% for the sectors transport, residential, energy, industry respectively. Especially the contribution of road transport is relevant, as its emissions are close to the ground in densely populated areas. Traffic information and population density will be used as proxies for urban emission (see Section 3.2.1 and 3.2.2).

In contrast to the regional atmosphere, the urban atmosphere is more dominated by dispersion processes, while many chemical

reactions are less important due to a relatively short residence time (Harrison, 2018). For the dispersion of the emission sources, the open source steady-state plume model AERMOD (Cimorelli et al., 2004) is used, developed by the American Meteorological Society (AMS) and United States Environmental Protection Agency (EPA). Based on the emission inventory and meteorology (see Section 3.2.4), AERMOD calculates hourly concentrations of air pollutants. The concentration distribution of an emission source is assumed to be Gaussian both horizontally as vertically when boundary layer conditions

are stable. In a convective boundary layer, the vertical distribution is described by a bi-Gaussian probability density function. Note that any other dispersion model can be used in the Retina methodology, as long as it is capable of simulating concentrations from individual emission sectors on an arbitrary receptor mesh.

### 3.1 AERMOD simulation settings

AERMOD version 16216r is used with simulation settings summarized in Table 1, operating on a rectangular domain of $18 \times$

22 km$^2$ covering the municipality of Amsterdam for the most part. All coordinates are reprojected in a custom oblique stereographic projection (EPSG:9809) around the city centre coordinate, such that the coordinate system can be considered equidistant at the urban scale. Instead of using a regular grid, a road-following mesh (Lefebre et al., 2011) is used. This reduces the number of receptor points, while maintaining accurate description of strong gradients found close to roads. Receptor locations are chosen at every 75 m along the parallel curves with 25 m distance to the road, and at every 125 m along the

parallel curves with 50 m distance to the road. The open spaces between these points are filled with a regular grid at 125 m resolution. Roads are modelled as line sources, while residential emissions are described as area sources. The dispersion is calculated for $NO_X$ to avoid a detailed analysis of the rapid cycling between its constituents NO and $NO_2$. Afterwards, an $NO_2/NO_X$ ratio is applied, depending on the available ozone ($O_3$), see Section 3.1.1.

Memory usage of AERMOD for the Amsterdam domain is proportional to the total number of emission source elements (here 17,069 road fragments and 12,182 residential squares) and the number of receptor points in the road-following mesh (here 42,128). The calculation time for a single concentration field is around 10 minutes, but can be reduced to a fraction of this by parallelizing the code.

### 3.1.1 Ozone chemistry and lifetime

Primary emissions of $NO_2$ (e.g. directly from the tailpipe) are only 5-10% of the total emitted $NO_X$ (Sokhi et al., 2007). At short time scales, secondary $NO_2$ is formed by oxidation of NO with $O_3$, while this reaction is counterbalanced by photolysis converting $NO_2$ to NO. The reaction rate of the first reaction is temperature dependent, while the latter depends on the available sunlight. The $NO_2/NO_X$ ratio has therefore an intricate dependence on temperature, radiation, and the proximity to the source (i.e. the travel time of the air mass since emission).

A practical approach to estimate this ratio is the Ozone Limited Method (OLM), as described in EPA (2015). The method uses ambient $O_3$ to determine how much NO is converted to $NO_2$. The dispersed (locally produced) $NO_X$ concentration is divided into two components: the primary emitted $NO_2$ (here assumed to be 10%), and the remaining $NO_X$ which is assumed to be all NO available for reaction with ambient $O_3$: $NO+O_3 \rightarrow NO_2+O_2$. If the mixing ratio of ozone ($O_3$) is larger than the 90% of ($NO_X$), than all NO is converted to $NO_2$. Otherwise, the amount of NO converted is equal to the available $O_3$, i.e. ($NO_2$) = $0.1(NO_X) + (O_3)$. The reaction is assumed to be instantaneous and irreversible. The resulting $NO_2$ concentration is added to the $NO_2$ background concentration.

Removal processes of $NO_X$ are modeled with an exponential decay. The chemical lifetime is in the order of a few hours. Liu et al. (2016) find $NO_X$ lifetimes in a range from 1.8 to 7.5 h using satellite observations over cities in China and the USA. Given the size of the domain and average wind speeds, its exact value is not of great importance here. Based on regression results a practical value of 2 hour is chosen.

## 3.2 Simulation input data

The dispersion simulation is driven by input data regarding emissions, background concentrations, and meteorology, listed in Table 2. All data, except for the traffic counts of inner city traffic, are taken from open data portals. The emission proxies are mapped in Fig. 2.

### 3.2.1 Traffic emissions

A recurrent problem when building urban air quality models is finding sufficiently detailed traffic emission information. Traffic emissions depend roughly on traffic flow and fleet composition, including engine technology. For many cities, unfortunately, this information is not available. Here a distinction is made between highways and primary roads, as both have a distinct traffic volume and weekly cycle. Differences in driving conditions and fleet composition are captured by assigning two different emission factors later on.

Road location data and road type definition data are taken from OpenStreetMap (OSM, 2017), which is a crowd-source project to create a free editable map of the world. A distinction is made between urban roads (labelled in OSM as "primary", "secondary", and "tertiary") and highways (labelled as "motorway" and "trunk"), as they have a distinct traffic pulse, fleet composition, and driving conditions. Road segments labelled as "tunnel" are not taken into account.

When the traffic flow $q$ (in vehicles per hour) is known, the emission rate $E$ for a road segment $l$ can be written as

$$E = \alpha_{\text{veh}} q l \tag{1}$$

with emission factor $\alpha_{\text{veh}}$ representing the (unknown) $NO_X$ emission per unit length per vehicle. Hourly traffic flow data is taken from 29 representative highway locations from the National Data Warehouse for Traffic Information (NDW, 2019), which contains both real-time and historic traffic data. For the urban traffic flow, data from 24 inductive loop counters provided by the traffic research department of Amsterdam municipality is used. Due to its large numbers, traffic flow is relatively well

predictable, especially when lower volumes during holiday periods and occasional road closures are neglected. For each counting site a traffic "climatology" is constructed, parametrized by hour and weekday, based on hourly data of 2016, see Fig. 3.

Traffic counts correlate strongly between different highway locations, all showing a strong commuting and weekend effect. Urban traffic typically shows, apart from lower volumes, less reduction between morning and evening rush hours, a less

pronounced weekend effect, and higher traffic intensities on Friday and Saturday night.

For locations $\mathbf{x}$ between the counting locations $\mathbf{x}_i$ the traffic flow $q(\mathbf{x})$ is spatially interpolated by inverse distance weighting (IDW):

$$q(\mathbf{x}) = \begin{cases} \frac{\sum_i w_i(\mathbf{x}) q_i}{\sum_i w_i(\mathbf{x})}, & \text{if } d(\mathbf{x}, \mathbf{x}_i) \neq 0 \text{ for all } i \\ q_i, & \text{if } d(\mathbf{x}, \mathbf{x}_i) = 0 \text{ for some } i \end{cases} \tag{2}$$

in which the weighting factors $w_i$ depend on the distance $d$ between $\mathbf{x}$ and the counting location $\mathbf{x}_i$:

$$w_i = \frac{1}{d(\mathbf{x}, \mathbf{x}_i)^2} \tag{3}$$

Validation in the Supplementary Material shows that for this counting network IDW predicts the traffic volume within a 50% error margin at most locations. Better results are obtained when more counting locations are available, or when they are selected strategically around crossings and access roads. Model simulations show that using inferior traffic data is partly compensated by the calibration (Section 4), at the expense of less pronounced concentration gradients.

**3.2.2 Population data**

Population density is considered to be a good proxy for residential emissions, e.g. from cooking and heating. Here, data is taken from the gridded population database of 2014, compiled by the national Central Bureau for Statistics (CBS, 2019) at a 100 m resolution. Each grid cell is offered to the dispersion model as a separate area source. To reflect the observation that residential emissions per capita are less when people are living closer to each other (Makido et al., 2012), the emission fluxes

are taken proportional to the square root of the population density $p$:

$$E = \alpha_{\text{pop}}\sqrt{p} \qquad\qquad\qquad\qquad (4)$$

### 3.2.3 Background concentrations

As AERMOD only describes the local contribution to air pollution, background concentrations are added which are taken from the Copernicus Atmosphere Monitoring Service (CAMS) European air quality ensemble (Marécal et al., 2015). The CAMS ensemble consists of 7 regional models producing hourly air quality and atmospheric composition forecasts on a $0.1 \times 0.1$ degree resolution. The analysis of the ensemble is based on the assimilation of up-to-date (UTD) air quality observations provided by the European Environment Agency (EEA). Each model has its own data assimilation system.

In the CAMS product the local contributions are already present. To get a better estimate for regional background concentrations avoiding double counts, the lowest concentration found in a $0.3 \times 0.3$ degree area around the city for $NO_2$ is taken, together with the mean concentration found in this area for $O_3$.

### 3.2.4 Meteorological data

The dispersion of air pollution is strongly governed by local meteorological parameters, especially the winds driving the horizontal advection and the characterization of the boundary layer which defines the vertical mixing. Meteorology also affects the chemical lifetime of pollutants.

AERMET (EPA, 2019) is used as a meteorological pre-processor for organizing available data into a format suitable for use by the AERMOD model. AERMET requires both surface and upper air meteorological data, but is designed to run with a minimum of observed meteorological parameters. Vertical profiles of wind speed, wind direction, turbulence, temperature, and temperature gradient are estimated using all available meteorological observations, and extrapolated using similarity (scaling) relationships where needed (EPA, 2018).

Hourly surface data from the nearby Schiphol airport weather station can be obtained from the Integrated Surface Database (ISD, see Smith et al. (2011)). Observations of temperature, winds, cloud cover, relative humidity, pressure, and precipitation are retrofit to match the SAMSON data format (WebMet, 2019a) which is supported by AERMET. Upper air observations are taken from daily radiosonde observations in De Bilt (at 35 km from Amsterdam), archived in the Integrated Global Radiosonde Archive (IGRA) (Durre et al., 2006). Pressure, geopotential height, temperature, relative humidity, dew point temperature, wind speed and direction are converted to the TD6201 data format (WebMet, 2019b) for each reported level up to 300 hPa.

### 4 Calibrating the model

Using proxy data instead of real emission introduces the problem to find the emission factors which best relate the activity data to their corresponding emissions. Instead of using theoretical values or values found in literature, effective values are derived which best fit the hourly averaged $NO_2$ observations of a network of $N$ stations.

For a certain hour $t$, the emission of a source element $i$ belonging to source sector $k$ can be written as

$$E_{ik}(t) = \alpha_k P_{ik}(t) \qquad\qquad (5)$$

in which $P_{ik}$ represents the corresponding emission proxy. The contribution of this source to the concentration at a receptor location $j$ is

$$c_{ijk}(t) = f_{ij}(t)E_{ik}(t) \qquad\qquad (6)$$

with $f_{ij}$ describing the dispersion of a unit emission from $i$ to $j$, including the conversion from $NO_X$ to $NO_2$ from the OLM. Eq. (6) is assumed to describe a linear relation between emission and concentration, although strictly speaking the variable $NO_2/NO_X$ ratio introduces a weak nonlinearity. A regression analysis is applied for a certain period, assuming that for each $t$ the total $NO_2$ concentration $c_j$ at station $j$ can be described as a background field $b$ and a local contribution consisting of a linear combination of the dispersed fields of $K$ emission sectors:

$$c_j(t) = b(t) + \sum_{k=1}^{K} a_k \sum_{i}^{S_k} f_{ij}(t)P_{ik}(t) \qquad\qquad (7)$$

$S_k$ represents the number of source elements for an emission sector $k$. The second sum in this equation is calculated for every hour with the Gaussian dispersion model taking the meteorological conditions during $t$ into account. Note that both background concentrations $b(t)$ and local concentrations $c_j(t)$ are taken from external data, see Section 3.2.3 and 2. Considering a period of $T$ hours, Eq. (7) can be interpreted as a matrix equation from which the emission factors $a_k$ can be solved using ordinary least

235 squares. Given the physical meaning of $a_k$, only positive regression results are allowed.

In this setup, the emissions are approximated by three sectors highway traffic, urban traffic, and population density ($K=3$). The resulting $a_k$ do not necessarily represent real emission factors. Their values partly compensate for unaccounted emission sectors and unrealistic modelling (e.g. based on wrong traffic data or an incorrect chemical lifetime). In Retina $a_k$ every 24 hours, based on observations of the preceding week ($T=168$). Doing so, the periodic calibration adjusts itself to seasonal cycles and

240 episodes not captured by the climatologies (e.g. cold spells or holiday periods). To avoid reducing the predictability of the regression model too much ($a_k$ dropping to zero), not all reference stations are used for calibration, but only stations classified as roadside or urban background. For the Amsterdam network, $N=11$. The residential emissions are represented by the population density, which is a time invariant proxy. To allow for a diurnal cycle, the residential emission factor is evaluated for two-hour bins. This brings the total number of fitted emission factors to 14: one for highway traffic, one for urban traffic,

and 12 describing the daily residential emission cycle.

Figure 4 shows an example of the air quality simulation after the emission factors have been determined. The stacked colours in the time series of Fig. 4b show that the contribution from different emission sectors to local air pollution can strongly vary from site to site.

**4.1 Diurnal and seasonal analysis of calibration results**

It is important to realize that the numerous modelling assumptions prevent that the calibration realistically solves the underdetermined inverse problem of finding the underlying $NO_X$ emissions based on the observed $NO_2$ concentrations. Instead, it evaluates how much $NO_X$ must be injected into the model to explain the observed spatial $NO_2$ patterns (unbiased with respect

to the calibration locations). To study the results of the regression analysis, a comparison was made between a summer month (July 2016, mean temperature 18.4 ºC) and a winter month (January 2017, mean temperature 1.6 ºC). Figure 5 shows the diurnal emissions for a 0.2 × 0.1 degree area, corresponding to the two grid cells of the EDGAR inventory covering the city centre.

Ideally, the emissions would be around the values found in the EDGAR inventory (6.23 and 7.18 $10^{-10}$ kg $NO_X/m^2/s$ for Summer and Winter respectively), and a corresponding ratio between residential and transport emissions (8% and 48% for Summer and Winter respectively). Unlike traffic, however, the diurnal cycle for the residential contribution is not prescribed, but is shaped in the regression analysis. The seasonal analysis shows that its fitted diurnal cycle not only describes changing residential emissions, but also compensates for changing $NO_2/NO_X$ ratios over the day (not included in the OLM) due to changing photochemistry and temperature. In daylight, the destruction of $NO_2$ by photolysis ($NO_2 + h\nu \rightarrow NO + O_3$) is strong, reducing the $NO_2/NO_X$ ratio. At low temperatures, the formation of $NO_2$ from NO ($NO + O_3 \rightarrow NO_2 + O_2$) is slow, also reducing the $NO_2/NO_X$ ratio. Also, due to collinearity, part of the traffic emissions will be explained by population density. Therefore, the found emission factors (and the corresponding sectoral emissions) should be considered as "effective" rather than real, i.e. as factors which best describe the observations under the given model assumptions.

## 5 Assimilation of observations

As the air quality network is spatially undersampling the urban area, the observations need to be combined with additional model information to preserve the fine local structures in air pollutant concentrations. The interpolation technique of choice here is Optimal Interpolation (OI) (Daley, 1991), having the desired property that the Bayesian approach allows for assimilation of heterogeneous measurements with different error bars. At an observation location the model value is corrected towards the observation, the innovation depending on the balance between the observation error and the simulation error. The error covariances determine how the simulation in the surroundings of this location is adjusted. Note that OI is essentially the same assimilation scheme as kriging-based approaches. The main advantage here is that one has detailed manual control over the error covariance matrix, which allows for a more comprehensive specification of the area of influence for each contributing observation. Outside the representativity range (i.e. the correlation length) of the observations, the analysis relaxes to the model values.

Consider a state vector $\mathbf{x}$ representing air pollutant concentrations on the (road-following) receptor mesh ($n\approx 40,000$). Define $\mathbf{x}^b$ as the background, i.e. the model simulation. Observation vector $\mathbf{z}$ contains $m$ measurements, typically 10☐ 100. Following the convention by Ide et al. (1997), the OI algorithm can now be written as:

$$\mathbf{x}^a = \mathbf{x}^b + \mathbf{K}(\mathbf{z} - H(\mathbf{x}^b)) \tag{8}$$

$$\mathbf{K} = \mathbf{P}^b \mathbf{H}^T (\mathbf{H} \mathbf{P}^b \mathbf{H}^T + \mathbf{R})^{-1} \tag{9}$$

$$\mathbf{P}^a = (\mathbf{I} - \mathbf{KH}) \mathbf{P}^b \tag{10}$$

Matrix $\mathbf{R}$ is the $m \times m$ observation error covariance matrix. As all observations are independent (the measurement errors are uncorrelated), $\mathbf{R}$ is a diagonal matrix with the measurement variances on its diagonal.

$\mathbf{P}^b$ is the $n \times n$ model error covariance matrix, describing how model errors are spatially correlated. The calculation of $\mathbf{P}^b$ is not straightforward; in Section 5.1 an approximation is derived.

Operator $H$ is the forward model, which maps the model state to the observed variables and locations. The matrix calculations can be simplified by reserving the first $m$ elements of the state vector for the observation locations, and the other $n - m$ elements for the road-following mesh. The Gaussian dispersion model is evaluated "in-situ" at the observation locations. Avoiding reprojection or interpolation means that there are no representation errors associated with $H$. The simulations at the observation locations $\mathbf{z}^b$ can then be written as a matrix multiplication

$$\mathbf{z}^b = H(\mathbf{x}^b) = \mathbf{H}\mathbf{x}^b \tag{11}$$

in which $\mathbf{H}$ is an $m \times n$ matrix for which its first $m$ columns form a unity matrix, while its remaining elements are 0.

Eq. (8) describes the analysis $\mathbf{x}^a$, i.e. how the observations $\mathbf{z}$ are combined (assimilated) with the model $\mathbf{x}^b$. It is a balance between the model covariance and the observation covariances, described by the gain matrix $\mathbf{K}$ in Eq. (9). $\mathbf{K}$ determines how strong the analysis must incline towards the observations or remain at the simulated values, to obtain the lowest analysis error variance, $\mathbf{P}^a$ in Eq. (10).

Note that Eq. (8)-(10) are analogous to the first step in Kalman filtering. The second step of the filter, propagating the analysis to the next time step, cannot be made here as the plume model solves a stationary state which is independent of the initial air pollutant concentration field. Also note that since an approximated model error covariance matrix will be used, generally these equations do not lead to an optimal analysis, hence this approach is more correctly referred to as Statistical Interpolation.

Let vector $\mathbf{c}$ represent the observed $NO_2$ mass concentrations, as described in Section 2. The distribution of the air pollutant concentrations resembles better the lognormal distribution than the Gaussian distribution, as can be seen from the Q-Q plots in Fig. 6. The analysis is done in log-space ($z_j = \ln c_j$), stabilizing the results by reducing the impact of less frequent measurements of high concentrations. Once returning from the log domain, Eq. (8) can be rewritten as:

$$\mathbf{c}^a = \exp(\mathbf{x}^a) = \mathbf{c}^b \exp(\mathbf{K}\Delta\mathbf{z}), \quad \text{with innovation vector } \Delta\mathbf{z} = \mathbf{z} - \mathbf{z}^b \tag{12}$$

By doing the analysis in the log-domain the assimilation updates correspond to multiplication instead of addition: $\exp(\mathbf{K}\Delta\mathbf{z})$ represents the local multiplication factor with which the simulated concentration $\mathbf{c}^b$ is corrected. This means that the shape of the model field (e.g. strong gradients found close to busy roads) is locally preserved. Note that the error in $z_j$ corresponds to the relative error in $c_j$ : $dz = d(\ln c)/dc = dc/c$ . The observation error covariance matrix is therefore $\mathbf{R} = \text{diag}(\sigma_1^2, \sigma_2^2, \dots, \sigma_m^2)$, with $\sigma_j$ the relative error corresponding to observation $j$.

### 4.1 Modelling the model error covariance matrix

For an optimal result in the data assimilation a realistic representation of the model covariance matrix $\mathbf{P}^b$ is essential. The model covariances influence the spatial representativity of the observations: when model errors correlate over larger distances, the assimilated observation will change the analysis over a longer range.

Tilloy et al. (2013) choose to model the covariances depending on the road network. Error correlations are assumed to be high on the same road or on connected roads. For background locations, the correlation decreases fast in the vicinity of a road, while the error correlation between two background locations remains significant across a larger distance. The error covariances are kept constant in time, and taken independent of traffic conditions.

However, $\mathbf{P}^b$ changes from hour to hour, mainly because varying meteorology changes the atmospheric dispersion properties. Here, the model error covariance is estimated for each hour based on the spatial coherence of the simulated concentration field. The covariance between two grid locations $\mathbf{x}_i$ and $\mathbf{x}_j$ can be expressed as their correlation $\rho$ and their standard deviations $\sigma$:

$$P_{ij}^b = \sigma_i \, \rho(\mathbf{x}_i, \mathbf{x}_j) \, \sigma_j \tag{13}$$

The model error $\sigma$ can only be evaluated at locations of the reference network using time series analysis. These model errors are spatially interpolated to other grid locations using IDW, analogous to Eq. (2)-(3). The correlation of model errors between different locations is parametrized with a downwind correlation length $L_{dw}$ and a crosswind correlation length $L_{cw}$. The extent of the correlation lengths reflect the turbulent diffusion and transport of the Gaussian dispersed plumes for a specific hour. From spatial analysis of the simulation data a heuristic model is derived which describes the dependence of the correlation on distance:

$$\rho(d) = \exp\left(-\sqrt{d}\right), \tag{14}$$

with $d$ the scaled distance between $\mathbf{x}_i$ and $\mathbf{x}_j$ (expressed as $x_{dw}$ and $x_{cw}$ along the downwind and crosswind axes)

$$d = \sqrt{\left(\frac{x_{dw}}{L_{dw}}\right)^2 + \left(\frac{x_{cw}}{L_{dw}}\right)^2}, \tag{15}$$

such that all points on an ellipse with semi-major axis $L_{dw}$ and semi-minor axis $L_{cw}$ have the same correlations.

To fit the parameters $L_{dw}$ and $L_{cw}$ for a certain hour, 1000 sample locations are selected from the road-following mesh. To represent both polluted and less polluted areas, the locations are selected such that their concentrations are homogeneously distributed over the value range, excluding the first and last 5 percentile. For this sample, correlation lengths $L_{dw}$ and $L_{cw}$ are fitted using Eq. (14) and (15).

Figure 7 shows the results of this analysis for two different hours. For fields with low gradients (e.g. when traffic contribution is low at night), large values of $L$ can occur. To prevent assimilation instabilities, the fitted values of $L$ are limited to a maximum of 10 km. During the 2016 summer months, longest correlation lengths are found for fields with low gradients. Average midnight values, when traffic contribution is low, are about 8 km. Correlation lengths are shortest during the morning rush hour (~1 km), increasing to 3 km during the late morning and afternoon. There is a wind dependency, as stronger winds stretch the pollution plumes, increasing correlation lengths. From the fit results the average ratio between $L_{cw}$ and $L_{dw}$ is found to be 68%.

Once the covariance parameters are known, the covariance matrix elements are calculated with Eq. (13). Note that for the calculation of the gain matrix $\mathbf{K}$ there is no need to calculate the full $\mathbf{P}^b$ matrix. Instead, $\mathbf{P}^b\mathbf{H}^T$ is calculated, which due to the structure of $\mathbf{H}$ this matrix product corresponds to the first $m$ columns of the $n \times n$ matrix $\mathbf{P}^b$.

## 6 Validation of simulation and assimilation

To assess the data quality across the domain, a leave-one-out analysis is performed at all locations of the reference network for the period June 1 - August 31, 2016. The results are summarized in Table 3. Figure 8 illustrates two examples; plots for all validation locations can be found in the Supplementary Material. For the observation-free simulation (i.e. the model forecast) an average RMSE is found of 13.6 μg/m$^3$ and correlation of 0.57. When assimilating observations, the average RMSE drops to 10.4 μg/m$^3$ while the correlation increases to 0.78. Strong systematic underestimations of the simulation (characterized by

a large negative bias) are observed at street locations NL49002, NL49007 and industrial locations NL49546, NL49704. These are most likely caused by unrealistic assumptions of local emissions of either traffic or industry. The strong positive bias found at NL49014, located in a city park separated from the nearby main road by a block of 4-storey buildings, might be explained by an incorrect simulation of air pollutants in the direct vicinity of these buildings.

The CAMS regional ensemble analysis compares well with the average of the urban background stations; the very low bias (-

0.1 μg/m$^3$) corresponds with the fact that data of these stations are used in its analysis. (Note that the CAMS values used here correspond to the Amsterdam grid cell, not the 3×3 minimum values used as background for the modelling.) On the other hand, it shows strong underestimations at street locations, as expected. It is here where the Retina simulation outperforms the low resolution results of CAMS.

From Table 3 can be seen that the relative error in the model forecast (defined as the ratio between the RMSE and the mean of

the observations) is around 58% on average. When assimilating, the error becomes dependent on the distance to the nearest observation locations. For sites having the nearest assimilated observation within 2 km distance, the average RMSE drops from 16.8 to 11.9 μg/m$^3$, corresponding to an average relative error of 39%. For sites where the nearest assimilated observation is further away than 2 km, the average RMSE drops from 10.8 to 9.1 μg/m$^3$, corresponding to an average relative error of 53%.

## 7 Added value of low-costs sensors

The previous analysis is purely based on high-quality reference measurements. In this section is explored whether the statistical interpolation scheme can be used to derive useful information of low-cost measurements, despite their lower accuracy.

This is done by testing different assimilation configurations during the Urban AirQ campaign, from 15 June to 15 August 2016 (see Section 2). The campaign targeted a central area with 4 reference stations and 14 low-cost sensors (See Figure 10a). Validation is done for 5 different assimilation scenarios (AS):

- AS1: Assimilation of all reference measurements (leave-one-out);
- AS2: Assimilation of measurements from 3 central reference sites (leave-one-out);
- AS3: Assimilation of low-cost data only;
- AS4: Assimilation of measurements from 3 central reference sites (leave-one-out) and all low-cost data;
- AS5: Assimilation of all reference measurements (leave-one-out) and all low-cost data.

The results are summarized in Figure 9. As expected, results deteriorate when the number of reference locations in the assimilation are reduced from 14 (AS1) to 3 (AS2). The correlation decreases at all 4 validation locations. At NL49012, the RMSE increases significantly due to a positive jump in the bias. The lower analysis with respect to the observations is due to the absence of assimilation of high values at nearby street location NL49002, which enlarges the influence of lower observations found at urban background location NL49019. At NL49019, located in the middle of 3 assimilation locations, the

RMSE does not change significantly. Apparently, the effect of assimilation of observations farther than the surrounding locations is small.

When only observations of 14 low-cost sensors are assimilated (AS3), instead of observations at 3 reference sites (AS2), there is a notable improvement visible in bias and RMSE at location NL49019. Here, the low-cost sensors are relatively nearby, the closest being sensor SD04 at 120 m distance. At the other validation locations, the low-cost sensor assimilation results in

similar RMSE (i.e. within 1 $\mu g/m^3$), a comparable bias, but a slightly lower correlation.

The results can be further improved if both reference and low-cost sensor data are included (AS4 and AS5). At NL49019, the RMSE drops to 5.1 $\mu g/m^3$ (compared to 7.8 $\mu g/m^3$ when no low-cost data are included) while the correlation increases to 0.89. Again, there is no significant difference between including the three surrounding reference locations and including all reference locations. Also at street location NL49017 and urban background location NL49003, the inclusion of low-cost sensor data

improves RMSE and correlation compared to assimilations with reference data only (AS4 vs. AS2, and AS5 vs. AS1). At location NL49012, the bias reduces considerably only when all reference data are included in the assimilation (AS1 and AS5). The different assimilation scenarios show that low-cost sensor data assimilation improves the results locally, even in absence of reference data. Generally, the best results are obtained when both reference data and low-cost data are included. Assimilation can reduce local model biases. However, unrealistically modelled covariances can lead locally to the introduction of an

additional bias.

Next, a monthly averaged concentration map of Amsterdam is constructed with all reference data and all low-cost sensor data from the first half of the Urban AirQ, see Fig. 10b. The addition of the low-cost data lowers the assimilation results by several $\mu g/m^3$ in the undersampled area west of Oude Schans (NL49019), while the $NO_2$ increases with several $\mu g/m^3$ around the traffic arteries found south and east of this location (Fig. 10c). A large fraction of traffic on these roads uses the IJ-tunnel to

cross the river. On a monthly basis, this tunnel is used by approximately one million vehicles.

The second half of the Urban AirQ campaign coincides with the start of the summer holiday period and the closure of the IJ-tunnel for maintenance. Comparison of the $NO_2$ concentration maps of both periods reveal interesting features (Fig. 11). Based on averaged NO2 measurements at rural stations NL49565 and NL49703, the NO2 reduction due to meteorological variability is estimated to be 7%. The overall drop in $NO_2$ concentrations in the central area, however, is around 10% due to reduced

traffic during the summer break. Notable exception is the historic city centre, where the $NO_2$ reduction is only a few percent, probably related to the steady economic activity driven by tourism. The strongest $NO_2$ reductions, around 15%, are found around the access ways of the IJ-tunnel. A few main roads (e.g. De Ruijterkade/Piet Heinkade and Ceintuurbaan) show less $NO_2$ reduction than average, apparently due to redirected traffic avoiding the tunnel.

## 8 Discussion and Conclusions

As air pollution gradients can be strong in the urban environment, it is essential to combine (sparse) measurements with an air quality model when aiming at street-level resolution. Retina is a practical approach to interpolating hourly urban air quality measurements. The first step consists of a simulation by a dispersion model which is driven by meteorological data and proxies for traffic and residential emissions. The model is daily calibrated with historic measurements. In the second step, observations of different accuracy are assimilated using a statistical interpolation scheme.

Validation analysis confirms that the European CAMS ensemble is a good predictor for hourly $NO_2$ concentrations found in the urban background. However, the CAMS data for $NO_2$ can be misleading when interpreted at the local scale, as the predicted diurnal cycle often deviates substantially from that observed at urban air quality stations. Local effects can be better resolved when CAMS data is used for background concentrations in a dispersion model which is driven by proxies for traffic and residential emissions.

The Retina simulation setup shows that such a system can be built from open software and open data. Applied to Summer 2016 in Amsterdam, it reduces the relative error at street locations from 70% to 51%, mainly by reducing the negative bias from 18.2 to 5.3 µg/m3. At urban background locations the dispersion model often introduces a positive bias, especially when traffic sources are nearby.

       The mapping results improve considerably with the second Retina step when available observations are assimilated by the
statistical interpolation scheme. When assimilating measurements of the reference network, the relative error in $NO_2$ concentrations drops to 44% on average. The local error depends on the distance to the nearest observations: approximately 39% within 2 km of an observation site, increasing to 53% for larger distances. The typical correlation increases from 0.6 to 0.8.

       The Bayesian assimilation scheme also allows us to improve the results by including low-cost sensor data, in order to get
improved localized information. However, biases must be removed beforehand with careful calibration, as most low-cost air quality sensors suffer from issues like cross-sensitivity or signal drift, see e.g. Mijling et al. (2018). The assimilation of low-cost sensor data from the Urban AirQ campaign reveals more detailed structure in concentration patterns in an area which is undersampled by the official network. The additional measurements correct for wrong assumptions in traffic emissions used in the apriori interpolation, and give better insight into how traffic rerouting (for instance due to closure of an arterial road)
affects local air quality.

       Retina has been built on open data to facilitate a flexible application to other cities. The meteorology needed for AERMOD is taken from global data sets of ISD and IGRA. Road network information can also be obtained globally from OpenStreetMap. Traffic data tends to be hard to obtain. When no local data is available on diurnal and weekly traffic flow its patterns should be estimated. In the absence of local census data, population density data can be taken from the Global Human Settlement
database (Schiavina et al., 2019), which has global coverage on a 250 m resolution. For application within Europe, the

necessary background pollutant concentrations can be obtained from CAMS. For applications outside Europe other data sets have to be found.

In general, degraded input data and imperfections in the dispersion modelling will deteriorate the system's capability to resolve local structures; it will lower the effective spatial resolution of the simulations. In its extreme it will only describe the blurry
urban background pollution contribution added to the rural background. Oppositely, with improved input data and atmospheric modelling, the effective resolution will improve, reducing local biases. This is the focus of future research.

Significant inaccuracies due to local emissions which are not described adequately by the proxies (e.g. from industry, port and airport activity) can be reduced by including these sources explicitly in the dispersion modelling. Small-scale structures provoked by the local built-up area will be better described by introducing the street canyon effect. The model will also benefit
from a more detailed traffic emission model, based on more counting locations and aggregated from shorter time intervals. Ideally, such an emission model takes local differences in fleet composition also into account. Finally, simulations will gain accuracy with a more realistic $NO_X$ chemistry, concerning the $NO_X$ chemical lifetime (influencing the plume length) and the $NO_2/NO_X$ ratio.

Overall, the error of the assimilation results depends on the accuracy of the air quality model, the number of assimilated
observations, the quality of observations, and the distance to the observation location. A reasonable approximation of the model covariance matrix is found by assuming the model covariance to be isotropic and by fitting correlation lengths along the downwind and crosswind axes for every hour. Finding a more realistic description of the model covariance matrix will better suppress the introduction of bias by the assimilation, and will be subject to future research.

For near-real time monitoring and forecasting of air quality the CAMS ensemble analysis must be changed for the ensemble
forecast. Instead of observation-based meteorology one should use data from local or global numerical weather prediction models e.g. from the National Centers for Environmental Prediction (the Global Forecast System, GFS; open data) or the European Centre for Medium-Range Weather Forecasts (ECMWF; not open data).

Apart from assessment of historic data such as in this study, Retina has been applied successfully for near-real time monitoring and forecasting of $NO_2$ in the cities of Amsterdam, Barcelona, and Madrid. Future work includes the implementation of other
cities inside and outside of Europe, and the application of Retina to other pollutants such as particulate matter.

**Data availability**

**Author contribution**

All work in this study was carried out by the author.

**Competing interests**

The author declares that he has no conflict of interest.

**Acknowledgements**

The author wishes to acknowledge the people behind the data sources used in this study, most notably GGD Amsterdam (reference measurements), volunteers of the Urban AirQ campaign (low-cost measurements), NDW and Amsterdam Traffic Research Department (traffic data), contributors to OpenStreetMaps (road location and classification), CAMS (background

concentrations), IGRA and ISD (meteorology), and CBS (population data). This research was supported by KNMI-DataLab and the EU-H2020 project AirQast.

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

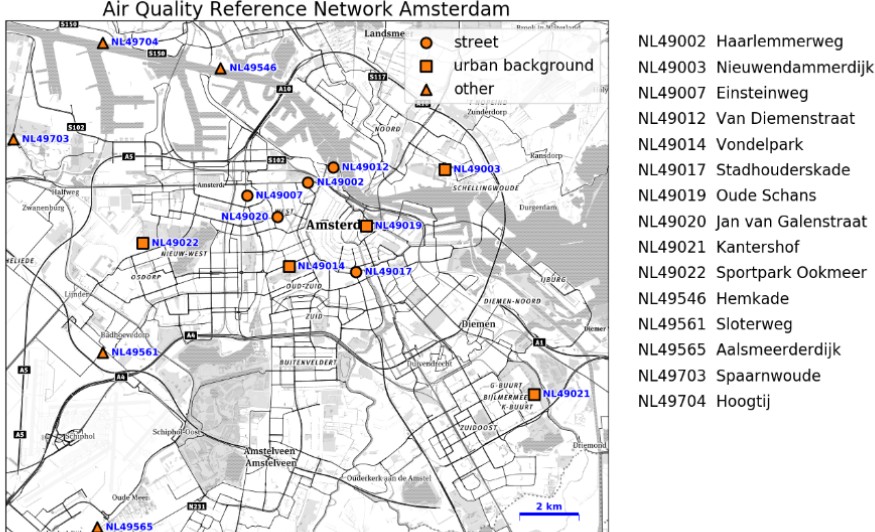

Air Quality Reference Network Amsterdam

| | |
|---|---|
| NL49002 | Haarlemmerweg |
| NL49003 | Nieuwendammerdijk |
| NL49007 | Einsteinweg |
| NL49012 | Van Diemenstraat |
| NL49014 | Vondelpark |
| NL49017 | Stadhouderskade |
| NL49019 | Oude Schans |
| NL49020 | Jan van Galenstraat |
| NL49021 | Kantershof |
| NL49022 | Sportpark Ookmeer |
| NL49546 | Hemkade |
| NL49561 | Sloterweg |
| NL49565 | Aalsmeerderdijk |
| NL49703 | Spaarnwoude |
| NL49704 | Hoogtij |

**Figure 1: Air quality reference network of Amsterdam. (Basemap source: © Mapbox © OpenStreetMap contributors 2019. Distributed under a Creative Commons BY-SA License.)**

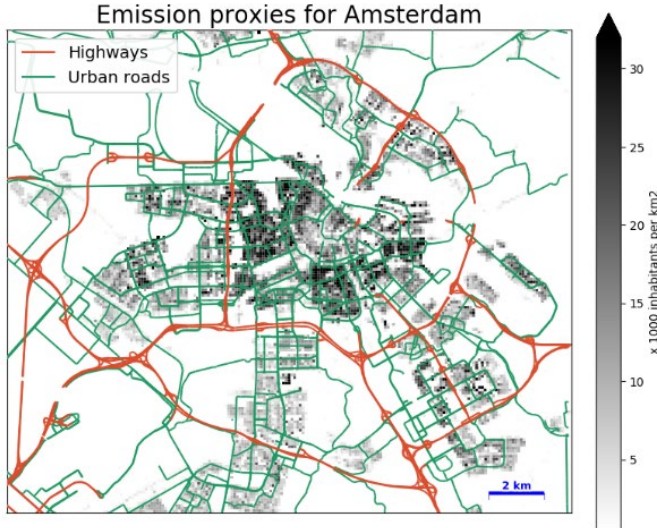

**Figure 2: Map of the emission proxies used for the dispersion model. Red lines indicate the highways, green lines indicate the urban**
**main roads. Grey colours indicate the population density. (Road location data adopted from © OpenStreetMap contributors 2019. Distributed under a Creative Commons BY-SA License.)**

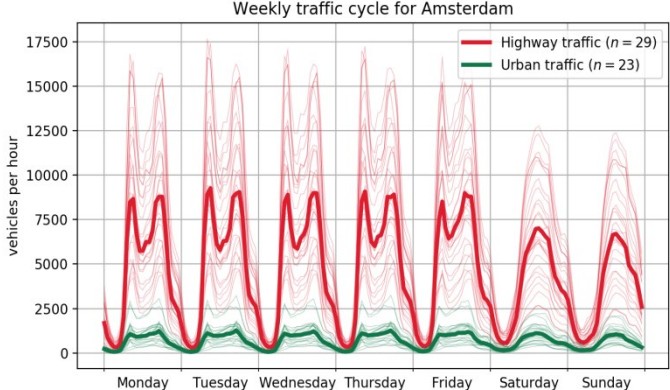

**Figure 3: Weekly cycle of highways and urban roads at counting locations (thin lines), aggregated from hourly data from 2016. The thick lines show the median of traffic flow for both road types. The morning and evening rush hours on working days are clearly visible for highways. Urban traffic has, apart from lower volume, less distinct peaks.**

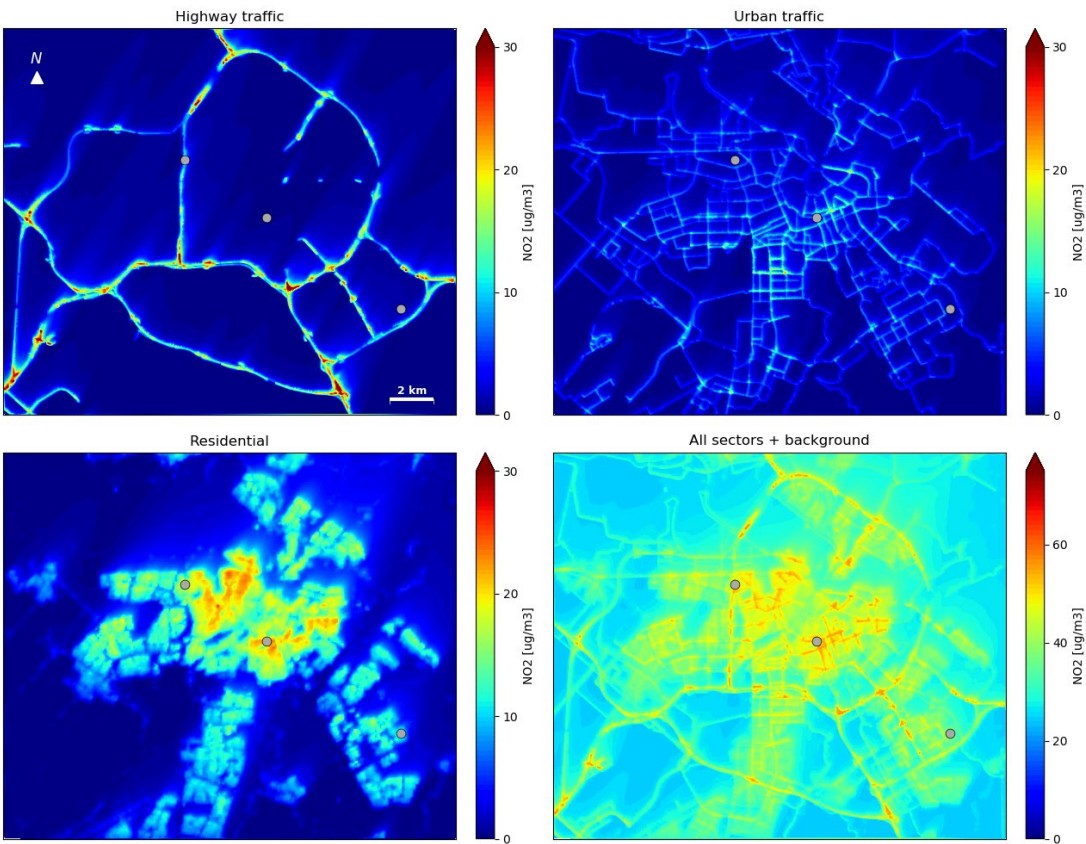

**Figure 4a: Dispersion maps of NO$_2$ concentrations for each emission sector at 8 July 2016, 9:00. The lower right panel shows the linear combination which best fits the time series at the calibration sites. Wind is blowing from the southwest at 16 km/h. The grey dots indicate an urban background location, a street location, and a highway location.**

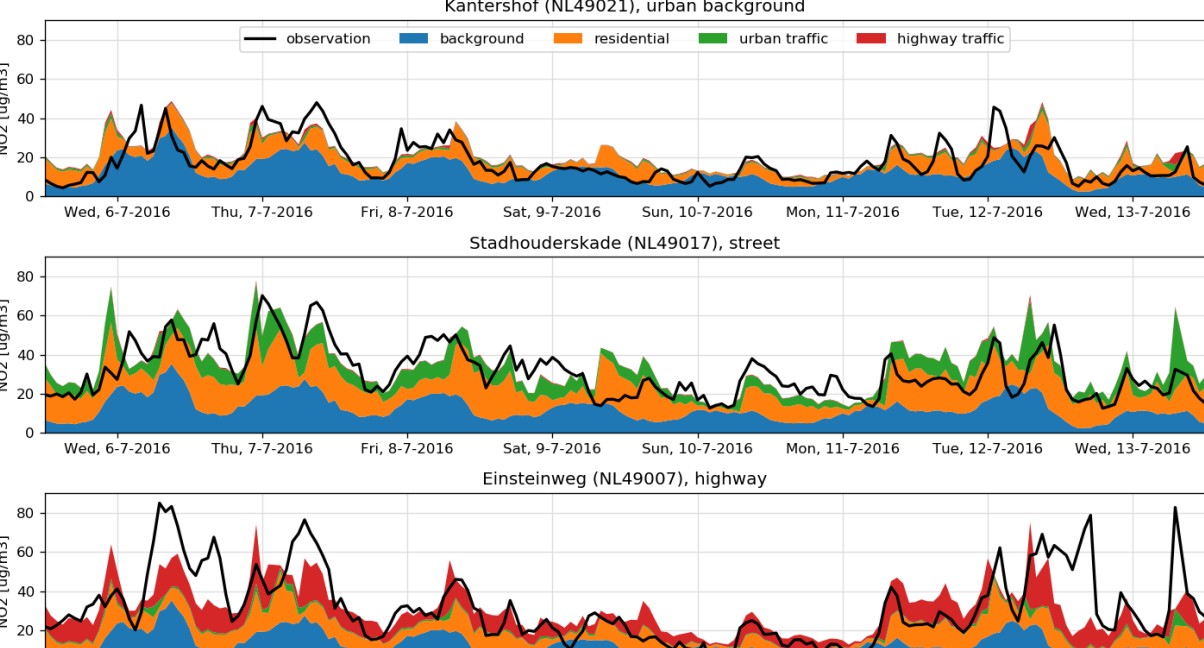

**Figure 4b: Comparison of observed and simulated NO₂ time series for the urban background location, the street location, and the highway location. The colours indicate the simulated contribution of the three source sectors and the background.**


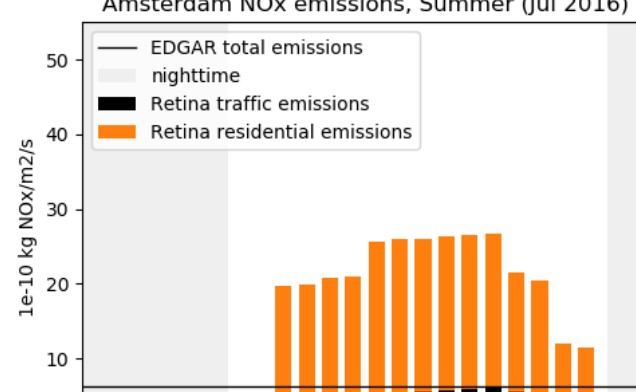
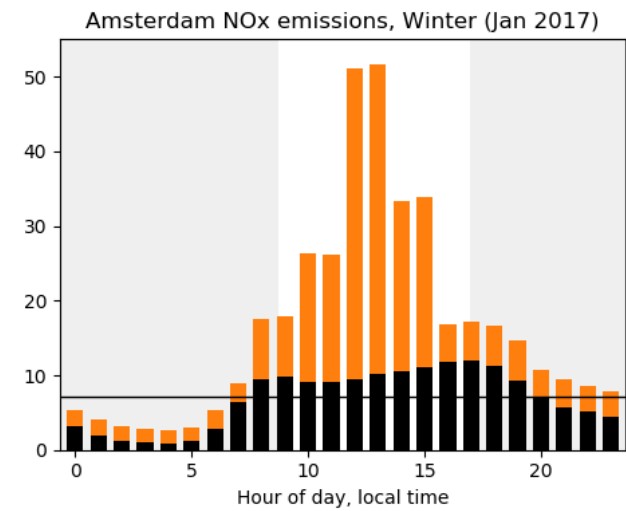

**Figure 5: Diurnal emission cycles after calibration of emission factors in different seasons.**

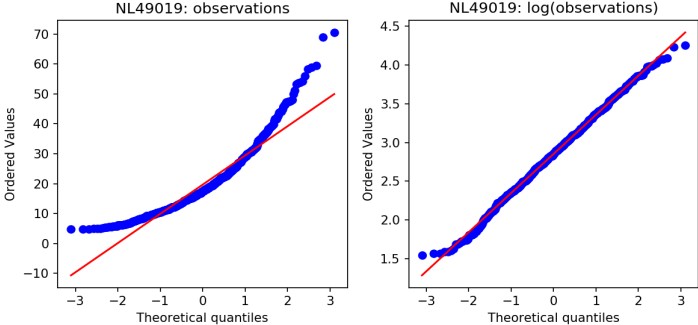

**Figure 6: (left) Distribution of the NO₂ observations at reference station Oude Schans in July 2016 compared to a standard normal distribution. (right) The logarithm of the observed values correspond better to a Gaussian distribution, shown by the quantile value pairs being almost on a straight line.**

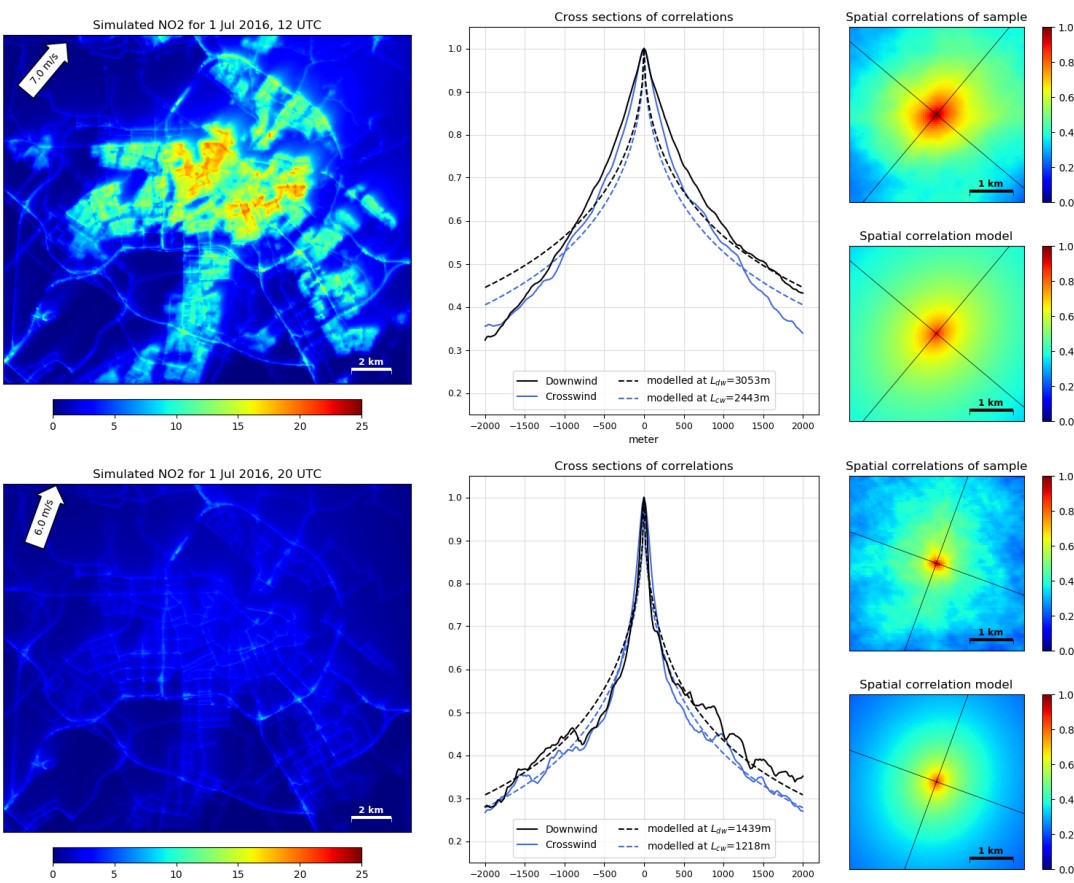


**Figure 7: Left panels show simulated NO₂ concentration fields at two different hours. The middle panels show the spatial correlations along the downwind and crosswind axes based on a sample of *n*=1000. The right panels show the spatial correlations of the sample and the resulting modelled spatial correlation model.**

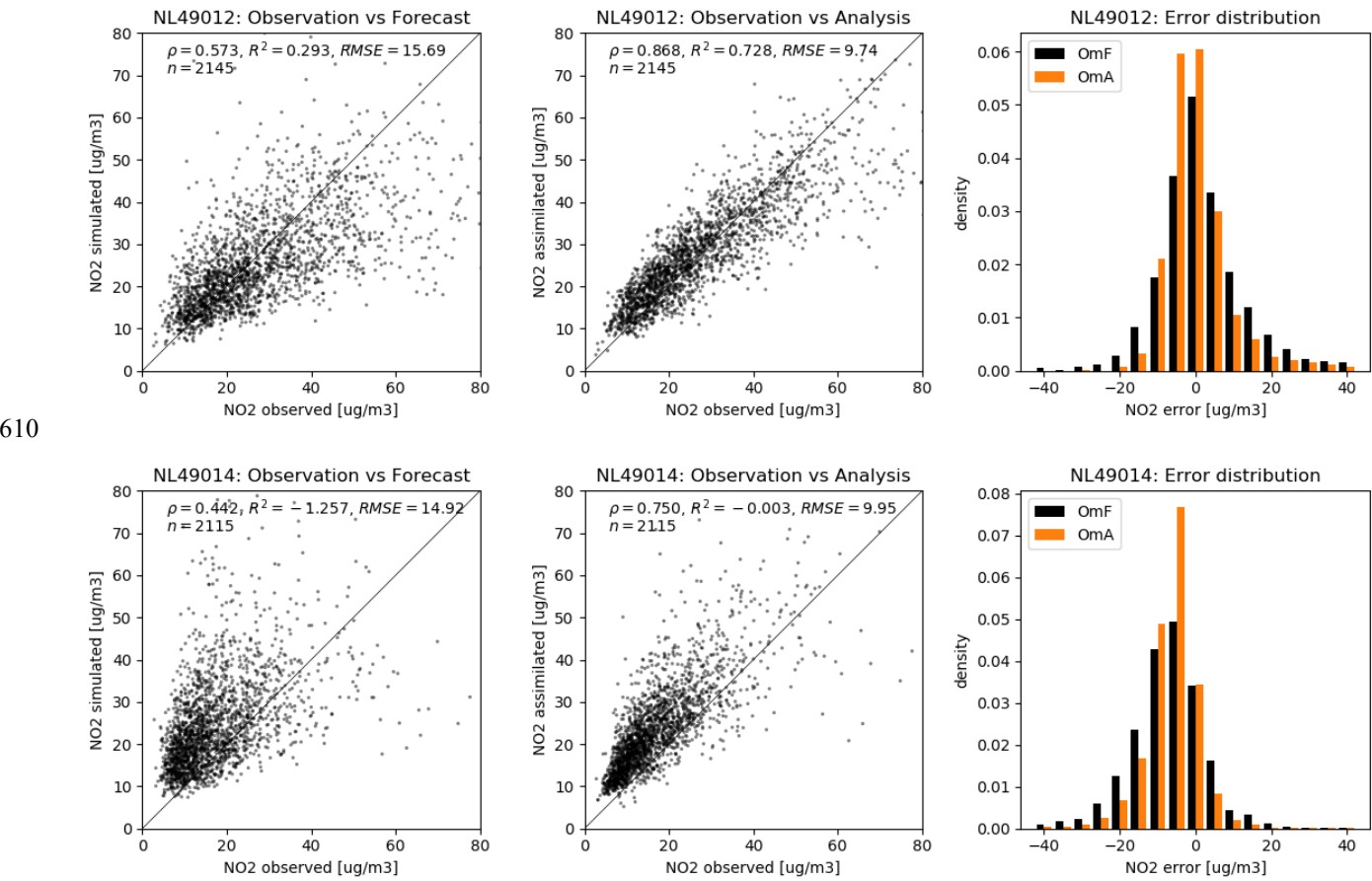

**Figure 8: Validation of hourly time series for the period 1 June to 31 August, 2016, for a well performing street location (above) and a less performing urban background location (below). Statistics of the n data pairs are given in correlation ($\rho$), coefficient of determination ($R^2$), and RMSE. The right hand panels compare the error distributions: the observation minus forecast (OmF) against the observation minus analysis (OmA).**

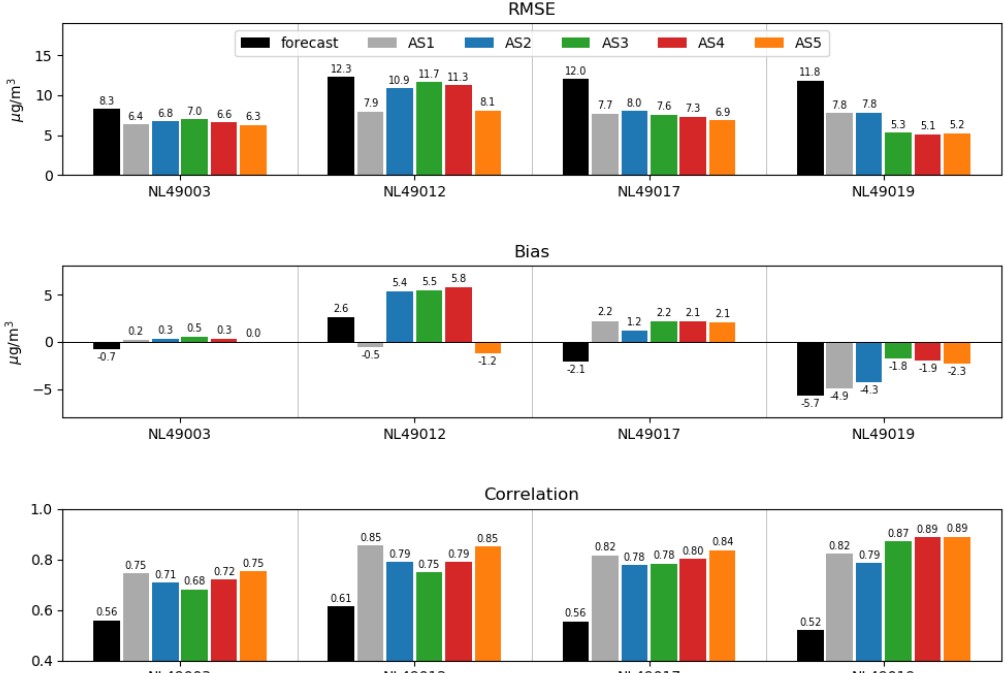

**Figure 9: Validation of the model forecast and five different assimilation scenarios at four central reference sites, for the period June 15 to August 15, 2016.**

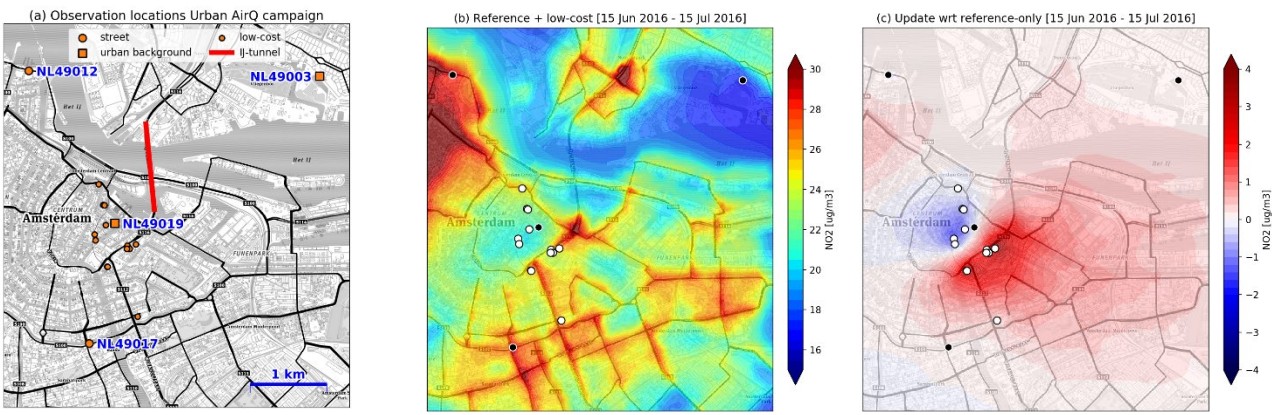

**Figure 10: (a) Observation sites during the Urban AirQ campaign. (b) 30-day average of NO₂ concentrations in the centre of Amsterdam, after assimilation of both reference measurements (black dots) and low-cost measurements (white dots). (c) Changes in spatial pattern when low-cost measurements are included in the analysis. (Basemap source: © Mapbox © OpenStreetMap contributors 2019. Distributed under a Creative Commons BY-SA License.)**

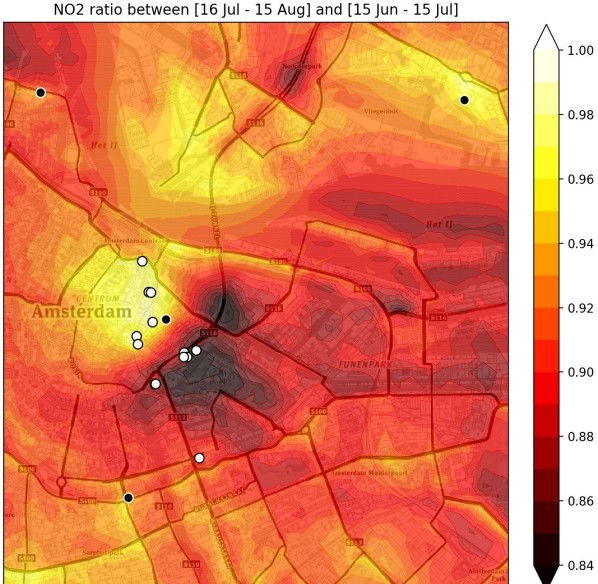

**Figure 11: Reduction of NO₂ during the holiday period. Largest reduction of concentrations is found in the vicinity of access ways to the IJ-tunnel, which was closed for maintenance. Concentrations in the historic centre remain unchanged. (Basemap source: © Mapbox © OpenStreetMap contributors 2019. Distributed under a Creative Commons BY-SA License.)**

**Table 1: Overview of AERMOD simulation settings**

| | |
|---|---|
| Road width | 20 m |
| Emission height traffic | 0.5 m |
| Emission height residential | 10 m |
| Initial vertical extension of concentration layer (sigma $Z_0$) | 10 m |
| Receptor grid | Road following |
| Receptor height | 1.5 m |
| Urban surface roughness length | 1 m |
| $NO_2/NO_X$ ratio | Ozone Limited Method (OLM), Primary emission ratio 10% |
| $NO_X$ lifetime | 2 h |
| Other AERMOD modelling options | Optimizing model runtime for sources (FASTALL) Address low wind speed conditions (LOWWIND3) Assuming flat terrain (FLAT) |

**Table 2: Summary of simulation input data for Amsterdam**

| Emission | Highway locations | OpenStreetMap (OSM, 2017): street segments labelled motorway and trunk |
| --- | --- | --- |
| | Urban road locations | OpenStreetMap (OSM, 2017): street segments labelled primary, secondary, tertiary |
| | Highway traffic flow | National Data Warehouse for Traffic Information (NDW, 2019): weekly cycle of vehicle counts at 29 selected locations (2016), interpolated to street segments |
| | Urban traffic flow | Amsterdam municipality (personal communication): weekly cycle of vehicle counts at 24 locations (2016), interpolated to street segments |
| | Population data | Statistics Netherlands (CBS, 2016): population density (2014) gridded at 100 m resolution |
| Observation | Background $NO_2$ | Copernicus Atmosphere Monitoring Service (CAMS, 2019): $NO_2$ analysis from model ensemble; minimum value found in 3x3 grid around domain centre |
| | Background $O_3$ | Copernicus Atmosphere Monitoring Service (CAMS, 2019): $O_3$ analysis from model ensemble; mean value found in 3x3 grid around domain centre |
| Meteorology | Meteorology (surface) | Integrated Surface Database (ISD, 2019): hourly observations from Schiphol Airport weather station |
| | Meteorology (upper air) | Integrated Global Radiosonde Archive (IGRA, 2019): daily radio sounding at De Bilt (NL) |

**Table 3: Validation results at reference locations, June 1-August 31, 2016**

| ID | name | type | n[1] | mean obs. | CAMS ensemble | | | Model forecast | | | Assimilated observations | | | |
|----|------|------|------|-----------|---------------|-|-|----------------|-|-|--------------------------|-|-|-|
| | | | | | RMSE[2] | bias | corr | RMSE[2] | bias | corr | RMSE[2] | bias | corr | dist[3] |
| NL49002 | Amsterdam - Haarlemmerweg | street | 2145 | 42.2 | 31.4 | -25.6 | 0.49 | 22.6 | -14.3 | 0.55 | 18.6 | -14.5 | 0.83 | 0.99 |
| NL49007 | Amsterdam - Einsteinweg | street | 2145 | 38.1 | 29.2 | -21.4 | 0.42 | 19.6 | -6.9 | 0.57 | 16.5 | -6.2 | 0.72 | 1.26 |
| NL49012 | Amsterdam - Van Diemenstr. | street | 2145 | 29.1 | 20.2 | -12.5 | 0.53 | 15.7 | -2.7 | 0.57 | 9.7 | -0.5 | 0.87 | 0.99 |
| NL49017 | Amsterdam - Stadhouderskade | street | 2140 | 30.1 | 17.9 | -13.5 | 0.45 | 14.3 | 1.9 | 0.50 | 9.0 | -2.7 | 0.78 | 1.60 |
| NL49020 | Amsterdam - Jan van Galenstraat | street | 2131 | 34.8 | 24.0 | -18.2 | 0.59 | 16.6 | -4.7 | 0.58 | 11.1 | -5.3 | 0.86 | 1.26 |
| NL49003 | Amsterdam - Nieuwend. dijk | urban backgr. | 2145 | 16.6 | 8.6 | 0.1 | 0.60 | 10.5 | 2.0 | 0.47 | 7.5 | 0.8 | 0.71 | 3.28 |
| NL49014 | Amsterdam - Vondelpark | urban backgr. | 2115 | 17.3 | 9.0 | -0.7 | 0.52 | 14.9 | 7.9 | 0.44 | 9.9 | 6.5 | 0.75 | 1.73 |
| NL49019 | Amsterdam - Oude Schans | urban backgr. | 2124 | 20.7 | 10.3 | -4.1 | 0.59 | 13.8 | 5.8 | 0.50 | 8.7 | 4.6 | 0.81 | 1.60 |
| NL49021 | Amsterdam - Kantershof | urban backgr. | 2082 | 14.9 | 7.5 | 1.6 | 0.65 | 10.7 | 5.6 | 0.56 | 8.0 | 4.4 | 0.73 | 7.33 |
| NL49022 | Amsterdam - Sportp. Ookmeer | urban backgr. | 2124 | 14.3 | 8.4 | 2.4 | 0.65 | 9.2 | 3.4 | 0.66 | 8.0 | 3.7 | 0.80 | 3.89 |
| NL49565 | Oude Meer - Aalsmeerderdijk | rural | 2127 | 17.3 | 9.1 | -0.6 | 0.57 | 9.0 | -2.4 | 0.59 | 8.0 | -3.0 | 0.73 | 5.94 |
| NL49703 | Amsterdam - Spaarnwoude | rural | 2125 | 13.0 | 8.7 | 3.7 | 0.61 | 8.1 | 2.1 | 0.60 | 7.5 | 2.4 | 0.71 | 4.47 |
| NL49546 | Zaanstad - Hemkade | industry | 2145 | 22.9 | 14.3 | -6.2 | 0.63 | 15.0 | -8.1 | 0.66 | 13.0 | -8.3 | 0.83 | 3.26 |
| NL49704 | Zaanstad - Hoogtij | industry | 2120 | 19.6 | 12.7 | -3.0 | 0.66 | 13.4 | -6.0 | 0.72 | 12.1 | -6.4 | 0.84 | 3.72 |
| NL49561 | Badhoevedorp - Sloterweg | undecided | 2145 | 20.5 | 10.6 | -3.9 | 0.64 | 10.8 | -2.9 | 0.61 | 8.9 | -4.2 | 0.79 | 3.96 |
| **Average street locations** | | | | **34.9** | **24.5** | **-18.2** | **0.50** | **17.8** | **-5.3** | **0.55** | **13.0** | **-5.8** | **0.81** | |

| | | | | | | | | | | |
|---|---|---|---|---|---|---|---|---|---|---|
| Average urban background locations | 16.8 | 8.8 | -0.1 | 0.60 | 11.8 | 4.9 | 0.53 | 8.4 | 4.0 | 0.76 | |
| Average all locations | 23.4 | 14.8 | -6.8 | 0.57 | 13.6 | -1.3 | 0.57 | 10.4 | -1.9 | 0.78 | |

1) Number of samples
2) In units $\mu g/m^3$
3) The distance to the nearest observation site, in km