# Peer review of "High-resolution mapping of urban air quality with heterogeneous observations: a new methodology and its application to Amsterdam"

_Atmospheric Measurement Techniques, 2019_

## Referee Comment (RC1) · Anonymous Referee #1 · 13 Jan 2020

The manuscript describes an air pollution modelling and mapping framework for cities that combines two elements, a dispersion model and a measurement network. The dispersion model utilizes, among others, meteorological measurements, background pollutant concentrations from external modelling and three types of spatially distributed emission sources (highways, urban roads, residential). In principle, the framework is based on three steps that eventually lead to hourly concentration maps: First, the dispersion model is run. Second, the measurement network is used to calibrate the model, i.e. to estimate the emission factors of the three source types. Third, the measurements are assimilated to the model in order to locally improve the pollutant concentration field.

[Figure]

The manuscript is well written and clear. The underlying concept of the described framework convinces, especially its modularity. Many activities are going on that look for meaningful applications for low-cost sensors. Using such data for the assimilation in a dispersion model is an attractive option. However, some computation steps outlined in the manuscript are based on approximations and assumptions that are computationally fast but also simple. Also the input data used for the dispersion model are partly only proxies. The manuscript does not present in enough detail the consequences of these simplifications. I favor the publication of the manuscript in AMT after the following two main issues and the specific comments have been carefully addressed and the manuscript has been substantially revised.

1. Due to the modularity of the modelling framework it is necessary that the accuracy and limitations of the individual components are determined and described in more detail and a little bit more critical. For example, the used dispersion model is computationally fast but it seems that it has clear limitations when applied in an urban environment as e.g. building geometries are not an input. Similarly, the emissions used as input for the dispersion model are derived in a simple way that similar data is probably available also for many other cities but the accuracy of the derived emission inventory seems by far not to be optimal. I think that the gain in accuracy of a spatially variable pollutant field by assimilating measurement data strongly depends on the model's capability to resolve small scale structures. It should be made more clear in the manuscript if measurements adjust local deviations in emission source activity or only general model deficiencies. In the latter case the assimilation of measurements does not necessarily lead to throughout improved results in a local environment around the sensor.

2. The validation part should be extended. The model uses a proxy for residential emissions as input data. Residential emissions probably have a distinct seasonality. Therefore, I strongly recommend to use an additional winter period to validate the modelling framework and to analyse the resulting alpha_pop. Actually, the simulation

of a whole year would be best. No low-cost sensor data are required for this analysis.

Specific comments

- Introduction (section 1) or Setting up an urban air quality model (section 2): For completeness, a short discussion/list of other dispersion models that could be used as an alternative to AERMOD should be included.

- Traffic emissions (section 2.2.1): The author writes that traffic emissions are the dominant source of NO2 in the Amsterdam domain. Hence, traffic emissions are an important input factor for the simulation of the NO2 concentration. The interpolation of the vehicle counts for arbitrary locations based on the counting sites using IDW is practical but, at least for urban roads, possibly limitedly accurate as network characteristics are neglected. When I think of parallel roads in close distance, IDW would assign them similar vehicle counts, but in reality the true counts can be very different. An analysis/description/discussion of the accuracy of the resulting traffic input data should be added.

- Population data (section 2.2.2): The indication of the magnitude of different contributions (heating, cooking, others) to the total residential emissions in Amsterdam would be helpful. For the all-season applicability of the model: does the population database also include the spatial distribution of employees to account for heating emissions of office buildings? And for the selected period: Are heating emissions substantial in this summer period?

- Calibrating the model (section 3): First, lines 217 to 219 are unclear for me. After reading these sentences I was confused if $c_j(t)$ in Eq. 6 is the measured NOX concentration. But it is not, correct? Second, can you comment on how worse the model is performing when residential emissions are omitted? I guess that in the selected summer period heating emissions are nearly zero. Are the estimated two-hourly alpha_pop plausible and can you show that the temporal pattern of the values are related e.g. to cooking emissions? In Figure 3b, the contribution of residential emissions to the NO2

is surprisingly large given the fact that residential emissions are only 1/3 of the traffic emissions (stated in section 2).

- Assimilation of observations (section 4): The described algorithm is applied by using the pollutant concentrations transformed into the log-space. Here, one has to be aware that the distribution of the pollutant concentration at a particular location is not equal the measurement error that is required for the algorithm in this section. The measurement error is described in the manuscript as being dependent on the concentration (section 2.3). So, the reasoning of this transformation is not correct. However, I suppose that the transformation of the measurements into the log-space has a positive effect on the stability of the results as the modeling framework becomes less sensitive to (less frequent) measurements of high concentrations by reducing their impact. The transformation into the log-space is fine, but the respective paragraph should be reformulated.

- Modelling the model error covariance matrix (section 4.1): The interpolation of the model error by IDW might result in an error field that is too smooth in the urban environment given that the model is limitedly capable to represent small-scale structures (e.g. buildings). At least a comment should be included in the manuscript that points out this issue.

- Validation (section 5): The first paragraph is, as I understand, only an example where the modelling framework works well. It can be removed. Start with the second paragraph ("overall assessment"). Figure 6 can be presented directly after the overall assessment by discussing sites where the model performs well (NL49019) and where it performs less optimal (e.g. NL49002, NL49014). Figure 6 should include examples for both types. The time period of the used data in this Figure should correspond to table 3. Omit in the Figure the performance analysis of low-cost sensors but extend chapter 6. Moreover, add a file to the manuscript with supplementary materials where the scatter plots of the remaining, in the manuscript not presented air quality monitoring sites are shown analogue to Fig. 6 in order to provide the reader a clear picture of the model

performance.

- Added value of low-cost sensors (section 6): The material presented in this chapter is only qualitative. The two average concentration maps presented are not validated and so the accuracy of their differences is unclear. The single example of the "Oude Schans" site is not sufficient to show that low-cost sensors add value.

I have some questions here: What is the reasoning of largely adjusting the results of the dispersion model by low-cost sensors when there is also the option to improve the input data for the dispersion model? Is it possible to generate traffic input data for the dispersion model for each month based on the traffic counts you have access to? Moreover, NO2 concentrations also depend on meteorology. What fraction of the differences in the monthly aggregated concentration fields is related to different weather conditions?

As I understand the main benefit of the low-cost sensors for the modelling framework is the increased spatial resolution of the measurements. Here, I miss some sensitivity analyses or similar material regarding measurement network design. What options exist when using this modelling framework in reducing traditional air quality monitoring sites and adding low-cost sensors? The accuracy of low-cost sensors is reported to be about 30%. Is this enough for adding substantial information?

Technical comments

- page 4, lines 102-103. How are the parallel distances of 75 and 125 m related to the grid? Maybe reformulate this sentence to make it clearer.

- page 5, lines 143-144. Refers traffic "climatology" to counting sites?

- page 7, line 197. Mijling (2018) instead of Mijling (2017)?

- page 7, lines 204-205. Pik, Pki: keep consistent.

- page 7, lines 213-214. I would not say that b(t) is observed. It is rather the output of

another modelling system.

- page 9, line 266. Section 3 instead of Section 2.3?

- page 10, line 285-286. "Isotropic" is the wrong word here as it is not isotropic.

- page 11, line 336. Change to "lower accuracy".

- Figure 1. Add units to the x and y axes.

- Figure 2. What do the depicted lines show? Sample week, yearly aggregation? Add more information.

- Figure 3a. Add north arrow and scale in one of the four Figures. The meaning of the three dots should be explained already in the caption of Figure 3a.

- Figure 3b. Indicate in the Figure in an appropriate manner the weekday the dates refer to. 2016-07-07 –> Thu, 2016-07-07. In Figure 2 the distinct traffic pattern is shown. It is interesting if there is a clear relation between traffic and NO2 in the modelling results.

- Figure 5. Add units for x and y axes and scale in all Figures. Remove the first "and" in the second sentence of the caption.

- Figure 7. Add scale in both Figures. Moreover, the visibility of the points could be better in all the presented maps.

- Figure 8. Add location of IJ-tunnel and of the historic center.

- Table 3. Indicate more precisely the date period the analysed measurements refer to.

———————————————————

---

## Referee Comment (RC2) · Anonymous Referee #2 · 1 Feb 2020

**GENERAL COMMENTS**

The manuscript describes a relatively simple but quite portable urban-scale modelling system for air quality based on AERMOD, which has the capability of assimilating point-based observations using statistical interpolation. The system is demonstrated for the city of Amsterdam using observations of nitrogen dioxide from both the official reference stations as well as a low-cost sensor network. The topic of the manuscript is highly relevant and the described system is an elegant solution for better exploiting data from low-cost air quality sensor networks in an objective and meaningful fashion and to thus address the growing need of ever more detailed high-resolution maps of

urban air quality. Overall the manuscript is very well written and clearly structured, and I highly recommend it for publication once some of the issues listed below are adequately addressed.

One of my main concerns relates to the lack of detail in some of the sections, but particularly in the section that is supposed to demonstrate the added value from assimilating low-cost sensor data (Sec 6). Given that multiple previous studies have already successfully assimilated regular stations observations using OI (e.g. Tilloy et al. 2013), one of the more novel aspects of this study is the assimilation of low-cost air quality sensors. The manuscript goes through great lengths of building up a dispersion model system with simplified emissions as well as an OI assimilation scheme, but the added value from low-cost sensor networks is covered in just a few lines towards the end without much detailed analysis. I think the manuscript could be a lot stronger and have more impact if a more comprehensive analysis of this were carried out in Section 6.

Secondly, Section 2.2.1 on traffic emissions (which are crucially controlling NOx/NO2) left me scratching my head at times. I realize that the system is designed to be portable and thus the necessary input data should be kept to a minimum, but I wonder if some of the simplifications taken here are defensible. In particular spatial interpolation of traffic monitoring sites seems to me a quite crude approximation that introduces significant uncertainties in the modelling. Further, what about distinguishing different types of vehicles? Regular cars versus heavy trucks? Euro 4/5/6 emissions categories? These things can have a very significant impact on the modelling results for NO2 and I wonder if some more care in setting up the modelling would not be beneficial in the long run? This is particularly a concern in the sense that OI should technically only be used when model and observations are unbiased against each other and ignoring certain high polluting vehicle classes could introduce potentially damaging biases. At the very least, the author should discuss these potential issues and lay out future steps to resolve these problems. In the best case scenario, it would be good to see some sensitivity studies testing the modelled NO2 sensitivity to inclusion of these different classes.

Thirdly, I feel that the manuscript could benefit from some more detail on how the error characteristics of the observations were derived. Estimating uncertainties from reference instruments and low-cost sensors on its own is a difficult subject and the paper does not provide the reader with information on how these were estimated or how such uncertainties were then transformed into error characteristics suitable for ingestion in the OI scheme. Such a discussion should be included in the paper and I believe this would strengthen the authors conclusions.

Finally, given that 90% of the paper deals with modelling and data assimilation, I do find the choice of AMT for this manuscript slightly puzzling and I feel that the paper would probably be better suited for a journal more focused on modelling or general air quality issues. However since the editor has accepted the paper to AMTD I assume that the material is considered suitable for the journal.

DETAILED COMMENTS

L15: "Retina" - why is it called this? Include the full name if this is an acronym.

L16/17: how are these percentages to be interpreted? Would be good mention here how accuracy is defined. Something as simple as "... a typical accuracy (defined here as [...]) of 39%" or similar

L23: "enhanced understanding of reference measurements". Please

L38: "adding value". I suggest you give an example of what you consider as adding value to the measurements or otherwise better write "exploting the measurements"

L39: In single-author papers it looks quite odd to use plural terms such as "our" and "we". Consider revising.

L43: I would add here that it depends on the mapping resolution and the pollutant. The required sampling density increases with the desired spatial resolution of the map. Furthermore, NO2 with is very sharp spatial gradients will always require a much denser network than for example mapping PM2.5 with its relatively smooth spatial gradients.

L60: The introduction/background section is missing a reference and a discussion of Tilloy et al (2013) (https://agupubs.onlinelibrary.wiley.com/doi/full/10.1002/jgrd.50233), who have essentially done the same as this paper (OI of point-based observations into an urban-scale AQ model).

L61: Again, the name "Retina" comes a bit out of the blue. You should probably introduce here what the acronym stands for.

L70: I would be a bit careful with the term "calibration" in this context, given that it has a very specific meaning for measurements (both reference as well as sensors). Maybe reword or describe a bit more thoroughly what happens in this step.

L98: A reference to AERMOD would be useful here.

L99: "local equidistant coordinate system" - at this point you might as well give the actual projection you used. Presumably something UTM-like?

L100: "road-following grid". This is used as if this a commonly known term, which in my opinion it is not. So first of all you might want to introduce this term a bit more carefully by saying something like "we use a road-following grid, which is essentially....". Secondly, to me it sounds a bit weird to use the term "grid" in this context, when you are basically talking about a spatially irregular and scattered set of receptor points with higher density along road links. I think the term grid should be reserved for a somewhat regular arrangement of cells.

L148-152: I realize that the goal of this paper is not to build the world's best model so a certain amount of simplification is expected, but interpolating traffic flow using IDW seems to be an incredibly crude method. How can this method possibly work? Between two loop counters there will likely be many road segments that either have much more or much less traffic than at the observation sites, so I fail to understand how simply interpolating here can lead to useful results. I think this section needs more detail on how this is carried out and a robust demonstration that the chosen methods

are meaningful.

L164: Don't they assimilate UTD data? In that case it wouldn't be a day old but just a few hours (maybe better write "up to a day old" or so).

L230: This should be Figure 3b? Also, I think this Figure should be discussed a bit more (maybe in the discussion section?) for example with respect to potential reasons for the difference between model and observations, particularly for the highway location.

L235: I recommend to remove the term "geostatistical" here. While OI is mathematically very similar to kriging-based techniques (and it can in fact be shown that it provides identical results to kriging if the same inputs are used) it is not traditionally considered a part of the field of geostatistics. Geostatistics was developed in the mining and earth resources community (Matheron et al.), whereas OI was developed within the meteorological community (Gandin 1965).

L241: Again, I would not use the term "grid" for what is essentially a set of irregular, scattered receptor points.

L265: I think it would be good to mention here that Statistical Interpolation/OI is essentially the same assimilation scheme (just a different mathematical framework) as previous kriging-based approaches. The main advantage of OI over geostatistics (but also an added complexity) is that one has detailed manual control over the Pb covariance matrix, which allows for a more comprehensive specification of the the area of influence for each contributing observation.

L287: extend -> extent (or maybe magnitude?); also reflect -> reflects

L335-340: I think this section should be either left out entirely or expanded upon significantly. As it is currently it does not represent a robust demonstration that low-cost sensors add value to the system, since the effect has only been shown at a single site and not been analysed in detail. Demonstrating that the information from low-cost sensors can improve urban-scale air quality modelling is clearly a very worthwhile goal but

this short section reads unfortunately more like an afterthought than a proper analysis.

L354: I think it would also be worthwhile noting here that, while CAMS is definitely useful for providing background conditions and initial conditions, for NO2 the CAMS forecasts can be very misleading when interpreted at the local scale. The predicted diurnal cycle can often deviate substantially from that observed at urban AQ stations.

L356: Agreed. And in addition the higher resolution from a dispersion model is also much more appropriate than CAMS for applications such as exposure estimates etc.

L374: "Traffic data tend to be harder to obtain". That is very true (and maybe even an understatement) and is one of the most limiting factors in running local-scale dispersion models at random locations. Given that traffic is typically the most important source for NOx I have the suspicion that even the comparatively portable Retina methodology is likely to fail when no such traffic data is available at all. I think it would be worthwhile discussing here that at some point, if nearly all the crucial input data to Aermod is either of low quality or entirely missing, the resulting forecasted concentration fields will be so bad that any type of sophisticated data assimilation of observations is no longer very meaningful.

L396: It might be more detailed, but is it really much better? This section is too qualitative to draw much of a conclusion. As I said above, I think the manuscript would benefit from a more robust analysis along these lines.

Figure 2: The caption should indicate more clearly that the thin lines represent the traffic at individual stations.

Figure 3a: These maps would benefit from some basic cartographic elements, e.g. a background map from OpenStreetmap similarly to Figure 7/8, scale bar, coordinates etc.

Figure 5: "Units are in meters" - not all of them. I recommend to either label all axes properly or to have a more thorough caption describing the various elements of this

busy Figure in more detail. It would also be helpful to have labels for each subplot (a, b, c..) so that they can be better referred to in the caption.

Figure 6: I think it is a bit confusing that the time series is only for 8 days, whereas the scatter plots show an entire month of data. Why not show the time series also for the entire period? If it is a visualization issue, it could be plotted over multiple rows. Similar to my earlier comments I also think that the analysis here would benefit from looking at more than just a single station.

Figure 8: "IJ-tunnel" Should be marked on the map since non-locals will not be familiar with this.

---

## Author Comment (AC1) · 22 Jun 2020

**Response to reviewer #1**

I want to thank the reviewer for the useful comments. It resulted in a revised manuscript which puts more emphasis on the added value of low-cost sensors by including results for different assimilation configurations. Also, a sensitivity study on traffic emissions have been added, and the accuracy and limitation of the method are better discussed. To put more emphasis on actual measurements governing the spatial interpolation, the subsection on observations is now put forward as an independent section. New figures have been added to better support the content. Results which are considered of importance but distracting from the main argument have been moved to Supplemental Material. The Discussion section and Conclusion section have been joined together and rewritten.

Below is my response. In blue the original comments, and in black the answers.

I favor the publication of the manuscript in AMT after the following two main issues and the specific comments have been carefully addressed and the manuscript has been substantially revised.

1. Due to the modularity of the modelling framework it is necessary that the accuracy and limitations of the individual components are determined and described in more detail and a little bit more critical. For example, the used dispersion model is computationally fast but it seems that it has clear limitations when applied in an urban environment as e.g. building geometries are not an input. Similarly, the emissions used as input for the dispersion model are derived in a simple way that similar data is probably available also for many other cities but the accuracy of the derived emission inventory seems by far not to be optimal. I think that the gain in accuracy of a spatially variable pollutant field by assimilating measurement data strongly depends on the model's capability to resolve small scale structures. It should be made more clear in the manuscript if measurements adjust local deviations in emission source activity or only general model deficiencies. In the latter case the assimilation of measurements does not necessarily lead to throughout improved results in a local environment around the sensor.

The revised manuscript elaborates on the comments above. The assimilation corrects both general model deficiencies and local deviations. The leave-one-out validation results show that in most cases the assimilation improves the accuracy, by reducing the local modelling bias, and improving the precision. This is better shown by including histograms of Observation minus Forecast (OmF) and Observations minus Analysis (OmA) for all validation locations. The potential shortcomings of the method are better discussed. Further reduction of the bias will be the subject of future studies, concentrating on (1) improving the modelling of traffic emissions, (2) including street canyon effects, (3) improving $NO_X$ chemistry (ratio $NO_2/NO_X$ and lifetime), (4) improving the modelling of the model covariance.

2. The validation part should be extended. The model uses a proxy for residential emissions as input data. Residential emissions probably have a distinct seasonality. Therefore, I strongly recommend to use an additional winter period to validate the modelling framework and to analyse the resulting alpha_pop. Actually, the simulation of a whole year would be best. No low-cost sensor data are required for this analysis.

I have added an analysis of the seasonality of the emission proxy calibration in Section 4.1. It compares a summer period (July 2016) with a winter period (January 2017). The results show that the average emissions do not agree well with the expected emissions from a bottom-up inventory. The regression analysis of the dynamic calibration finds the best linear combination of traffic proxies and residential proxies which explains the observations. Unlike traffic, the diurnal cycle for the residential contribution is shaped by the regression analysis. The seasonal analysis shows that the fitted diurnal cycle for the residential sector not only describes the cycle of the residential emissions, but also compensates for changing $NO_2/NO_X$ ratios over the day due to changing photochemistry and temperature. Also, due to collinearity, part of the traffic emissions can be explained by population density. Therefore, the found emission factors (and the corresponding sectoral emissions) should be considered as "effective" rather than real, i.e. factors which best describe the observations under the given model assumptions.

**Specific comments**
Introduction (section 1) or Setting up an urban air quality model (section 2): For completeness, a short discussion/list of other dispersion models that could be used as an alternative to AERMOD should be included.
Added to the end of the introduction in Section 3: "Note that any other dispersion model can be used in the Retina methodology, as long as it is capable of simulating concentrations from individual emission sectors on an arbitrary receptor mesh."

Traffic emissions (section 2.2.1): The author writes that traffic emissions are the dominant source of NO2 in the Amsterdam domain. Hence, traffic emissions are an important input factor for the simulation of the NO2 concentration. The interpolation of the vehicle counts for arbitrary locations based on the counting sites using IDW is practical but, at least for urban roads, possibly limitedly accurate as network characteristics are neglected. When I think of parallel roads in close distance, IDW would assign them similar vehicle counts, but in reality the true counts can be very different. An analysis/description/discussion of the accuracy of the resulting traffic input data should be added.
Indeed, the question remained how well this IDW interpolation describes the traffic flow differences found for nearby roads of the same road type. This is now assessed by two different approaches (presented in the Supplementary Material, and mentioned at the end of Section 3.2.1): a leave-one-out validation to study the error in local traffic flow estimations, and a concentration validation study of dispersion simulations done under different traffic scenarios. The results show that for this counting network IDW predicts the traffic volume within a 50% error margin at most locations. The model simulations show that using inferior traffic data is partly compensated by the calibration dynamics, at the expense of less pronounced concentration gradients.

Population data (section 2.2.2): The indication of the magnitude of different contributions (heating, cooking, others) to the total residential emissions in Amsterdam would be helpful. For the all-season applicability of the model: does the population database also include the spatial distribution of employees to account for heating emissions of office buildings? And for the selected period: Are heating emissions substantial in this summer period?
This is now assessed in newly added Section 4.1, where the diurnal cycles of sectoral emissions are analysed for summer and winter. There is no clear answer to the reviewer's questions, as the NOx emissions used by the model after calibration should be regarded as

"effective" rather than real, i.e. best describing the spatial concentration patterns under given model assumptions. The effective residential emissions contain an unquantified contribution which compensates for the simplified NOx chemistry assumptions.

Calibrating the model (section 3): First, lines 217 to 219 are unclear for me. After reading these sentences I was confused if cj(t) in Eq. 6 is the measured NOX concentration. But it is not, correct?

The confusion arises from the fact that $P$ represents proxies for $NO_x$ emissions, while $c$ represents the observed $NO_2$ concentrations. The conversion from $NO_x$ to $NO_2$ is implicitly done by dispersion $f$, using an $NO_2/NO_x$ ratio from the Ozone Limiting Method. In Gaussian dispersion modelling there is a linear relation between emission strength and concentration, but this linearity breaks down when conversion from $NO_x$ to $NO_2$ is included. This is now better formulated by eliminating lines 217 to 219 and changing the description of Eq. (6) to: "(...) with $f_{ij}$ describing the dispersion of a unit emission from $i$ to $j$, including the conversion from $NO_x$ to $NO_2$ from the OLM. Eq. (6) is assumed to describe a linear relation between emission and concentration, although strictly speaking the variable $NO_2/NO_x$ ratio introduces a weak nonlinearity."

Second, can you comment on how worse the model is performing when residential emissions are omitted? I guess that in the selected summer period heating emissions are nearly zero. Are the estimated two-hourly alpha_pop plausible and can you show that the temporal pattern of the values are related e.g. to cooking emissions? In Figure 3b, the contribution of residential emissions to the NO2 is surprisingly large given the fact that residential emissions are only 1/3 of the traffic emissions (stated in section 2).

This is now assessed in Section 4.1, where also the collinearity between the traffic and residential proxies is mentioned.

Assimilation of observations (section 4): The described algorithm is applied by using the pollutant concentrations transformed into the log-space. Here, one has to be aware that the distribution of the pollutant concentration at a particular location is not equal to the measurement error that is required for the algorithm in this section. The measurement error is described in the manuscript as being dependent on the concentration (section 2.3). So, the reasoning of this transformation is not correct. However, I suppose that the transformation of the measurements into the log-space has a positive effect on the stability of the results as the modeling framework becomes less sensitive to (less frequent) measurements of high concentrations by reducing their impact. The transformation into the log-space is fine, but the respective paragraph should be reformulated.

The reviewer is right. Changed "*The analysis is therefore done in log-space ($z_j = ln \, c_j$), which converts lognormal distributions to Gaussian, for which the Bayesian assumptions behind Equation 8-10 are valid.*" to "*The analysis is done in log-space ($z_j = ln \, c_j$), stabilizing the results by reducing the impact of less frequent measurements of high concentrations.*"

Modelling the model error covariance matrix (section 4.1): The interpolation of the model error by IDW might result in an error field that is too smooth in the urban environment given that the model is limitedly capable to represent small-scale structures (e.g. buildings). At least a comment should be included in the manuscript that points out this issue.

It is impossible to assess the model error at all locations and under all conditions when the "ground truth" is only available at 15 locations. In my opinion, interpolation of the model error gives a good first impression of the local model performance away from validation locations. However, the reviewer is right that small-scale structures provoked by the local built-up area might not be well represented in the model. Inclusion of the street canyon effect, together with refinements in covariance modelling, is subject of further study. It will reduce local bias in the modelling, but also suppress the introduction of bias by the assimilation. This is now pointed out in the discussion section.

Validation (section 5): The first paragraph is, as I understand, only an example where the modelling framework works well. It can be removed. Start with the second paragraph ("overall assessment"). Figure 6 can be presented directly after the overall assessment by discussing sites where the model performs well (NL49019) and where it performs less optimal (e.g. NL49002, NL49014). Figure 6 should include examples for both types. The time period of the used data in this Figure should correspond to table 3. Omit in the Figure the performance analysis of low-cost sensors but extend chapter 6. Moreover, add a file to the manuscript with supplementary materials where the scatter plots of the remaining, in the manuscript not presented air quality monitoring sites are shown analogue to Fig. 6 in order to provide the reader a clear picture of the model performance.

The reviewer's suggestions have been implemented. The first paragraph has been deleted. Figure 6 has been replaced by two validation examples, for a well performing location (NL49012) and a worse performing location (NL49014). The time series plots have been removed, regarded as redundant as the performance can also be read from the scatter plots. Bar plots with error distributions have been added to better illustrate the effect of the assimilation. The validation plots for all reference locations are included in the Supplementary Material.

Added value of low-cost sensors (section 6): The material presented in this chapter is only qualitative. The two average concentration maps presented are not validated and so the accuracy of their differences is unclear. The single example of the "Oude Schans" site is not sufficient to show that low-cost sensors add value. I have some questions here: What is the reasoning of largely adjusting the results of the dispersion model by low-cost sensors when there is also the option to improve the input data for the dispersion model?

To start with the last question: better input data is not always available. In this particular case there was no detailed information available about traffic flow and changing traffic patterns. To better assess the added value of low-cost sensors, additional results have been added from different assimilation configurations. The different assimilation scenarios show that low-cost sensor data assimilation improves the results locally, even in absence of reference data. Generally, the best results are obtained when both reference data and low-cost data are included. This is the configuration used to obtain the map showing the $NO_2$ reduction during the holiday period (Figure 11).

Is it possible to generate traffic input data for the dispersion model for each month based on the traffic counts you have access to?

Yes, it would be possible to generate monthly traffic data. This is done for instance in traffic scenario TS3, which can be found in the Supplementary Material. However, as both the holiday period and the closure of the tunnel started at half July and ended half August, neither traffic data for July or August would be representative for the whole month.

Anticipating the application for other cities where such detailed traffic data might not be available, it was decided to evaluate the system for a yearly averaged traffic "climatology".

Moreover, NO2 concentrations also depend on meteorology. What fraction of the differences in the monthly aggregated concentration fields is related to different weather conditions?
The influence of meteorological variability is now included in the discussion of the figure. It can be estimated from the $NO_2$ reduction found at rural stations NL49565 and NL49703. Average values drop from 16.60 and 12.57 ug/m$^3$ during 15 June - 15 July, to 15.53 and 11.55 ug/m$^3$ during 16 July - 15 August. Added to this section: "*Based on averaged NO$_2$ measurements at rural stations NL49565 and NL49703, the NO$_2$ reduction due to meteorological variability is estimated to be 7%.*"

As I understand the main benefit of the low-cost sensors for the modelling framework is the increased spatial resolution of the measurements. Here, I miss some sensitivity analyses or similar material regarding measurement network design. What options exist when using this modelling framework in reducing traditional air quality monitoring sites and adding low-cost sensors? The accuracy of low-cost sensors is reported to be about 30%. Is this enough for adding substantial information?
These questions are now implicitly answered by the results of the different assimilation scenarios. Validation results are comparable when only observations of 14 low-cost sensors are assimilated (AS3), instead of observations at 3 reference sites (AS2). Even a notable improvement is visible in bias and RMSE at location NL49019, where the low-cost sensors are relatively nearby.

**Technical comments**
page 4, lines 102-103. How are the parallel distances of 75 and 125 m related to the grid? Maybe reformulate this sentence to make it clearer.
Changed to: "Receptor locations are chosen at every 75 m along the parallel curves with 25 m distance to the road, and at every 125 m along the parallel curves with 50 m distance to the road."

page 5, lines 143-144. Refers traffic "climatology" to counting sites?
Changed "For each location" to "For each counting site".

page 7, line 197. Mijling (2018) instead of Mijling (2017)?
Changed to Mijling (2018)

page 7, lines 204-205. Pik, Pki: keep consistent.
Adapted

page 7, lines 213-214. I would not say that b(t) is observed. It is rather the output to another modelling system.
Agreed, changed to "*Note that both background concentrations b(t) and local concentrations c$_i$(t) are taken from external data*"

page 9, line 266. Section 3 instead of Section 2.3?
**c** refers to the measurements of the reference network, described (according to the new numbering) in Section 2.

page 10, line 285-286. "Isotropic" is the wrong word here as it is not isotropic.
True. Changed to "*The correlation of model errors between different locations is parametrized with a downwind correlation length $L_{dw}$ and a crosswind correlation length $L_{cw}$.*"

page 11, line 336. Change to "lower accuracy".
Changed accordingly

Figure 1. Add units to the x and y axes.
Units are mentioned in the figure caption.

Figure 2. What do the depicted lines show? Sample week, yearly aggregation? Add more information.
Caption changed to "*Weekly cycle of highways and urban roads at counting locations, aggregated from hourly data from 2016. The morning and evening rush hours on working days are clearly visible for highways. Urban traffic has, apart from lower volume, less distinct peaks. The thick lines show the median of traffic flow for both road types.*"

Figure 3a. Add north arrow and scale in one of the four Figures. The meaning of the three dots should be explained already in the caption of Figure 3a.
Scale bar and arrow added. The figure caption now explains the three dots.

Figure 3b. Indicate in the Figure in an appropriate manner the weekday the dates refer to. 2016-07-07 –> Thu, 2016-07-07. In Figure 2 the distinct traffic pattern is shown. It is interesting if there is a clear relation between traffic and NO2 in the modelling results.
Labels on x-axis of Figure 3b changed.

Figure 5. Add units for x and y axes and scale in all Figures. Remove the first "and" in the second sentence of the caption.
Figure updated, scale bars added.

Figure 7. Add scale in both Figures. Moreover, the visibility of the points could be better in all the presented maps.
Figure 8. Add location of IJ-tunnel and of the historic center.
A new panel has been added showing the location of reference stations and low-cost sensors, the location of the IJ-tunnel, and the scale.

Table 3. Indicate more precisely the date period the analysed measurements refer to.
Table caption changed to "*Validation results at reference locations, June 1 - August 31, 2016*"

---

## Author Comment (AC2) · 22 Jun 2020

**Response to reviewer #2**

I want to thank the reviewer for the useful comments. It resulted in a revised manuscript which puts more emphasis on the added value of low-cost sensors by including results for different assimilation configurations. Also, a sensitivity study on traffic emissions have been added, and the accuracy and limitation of the method are better discussed. To put more emphasis on actual measurements governing the spatial interpolation, the subsection on observations is now put forward as an independent section. New figures have been added to better support the content. Results which are considered of importance but distracting from the main argument have been moved to Supplemental Material. The Discussion section and Conclusion section have been joined together and rewritten.

Below is my response. In blue the original comments, and in black the answers.

One of my main concerns relates to the lack of detail in some of the sections, but particularly in the section that is supposed to demonstrate the added value from assimilating low-cost sensor data (Sec 6). Given that multiple previous studies have already successfully assimilated regular stations observations using OI (e.g. Tilloy et al. 2013), one of the more novel aspects of this study is the assimilation of low-cost air quality sensors. The manuscript goes through great lengths of building up a dispersion model system with simplified emissions as well as an OI assimilation scheme, but the added value from low-cost sensor networks is covered in just a few lines towards the end without much detailed analysis. I think the manuscript could be a lot stronger and have more impact if a more comprehensive analysis of this were carried out in Section 6.

The work of Tilloy et al. is now referenced in the manuscript. This work is different in the sense that it presents a flexible urban dispersion model, it presents an alternative covariance model, and it studies the added value of low-cost sensors (details below). The section on the added value of low-cost sensor data has been extended with a sensitivity study of different network configurations.

Secondly, Section 2.2.1 on traffic emissions (which are crucially controlling NOx/NO2) left me scratching my head at times. I realize that the system is designed to be portable and thus the necessary input data should be kept to a minimum, but I wonder if some of the simplifications taken here are defensible. In particular spatial interpolation of traffic monitoring sites seems to me a quite crude approximation that introduces significant uncertainties in the modelling.

Spatial interpolation of traffic flow might indeed seem a crude way of solving the lack of traffic data. However, part of the introduced inaccuracy is avoided by distinguishing between two road types, highway and urban roads. Both having different diurnal patterns and vehicle counts, the traffic flow interpolation of one road type does not affect the other road type. The validity of the approach is now assessed in two different manners, presented in the Supplementary Material, and mentioned at the end of Section 3.2.1: a leave-one-out validation to study the error in local traffic flow estimations, and a concentration validation study of dispersion simulations done under different traffic scenarios. The results show that for this counting network IDW predicts the traffic volume within a 50% error margin at most

locations. The model simulations show that using inferior traffic data is partly compensated by the calibration dynamics, at the expense of less pronounced concentration gradients.

Further, what about distinguishing different types of vehicles? Regular cars versus heavy trucks? Euro 4/5/6 emissions categories? These things can have a very significant impact on the modelling results for NO2 and I wonder if some more care in setting up the modelling would not be beneficial in the long run? This is particularly a concern in the sense that OI should technically only be used when model and observations are unbiased against each other and ignoring certain high polluting vehicle classes could introduce potentially damaging biases.

The reviewer is right. We consider only two emission factors, one for highway traffic and one for urban traffic. The emission factors, however, are estimated from analysis against observations in the calibration phase, which implicitly compensates for i.e. different fleet composition. Further refinement of the traffic model is desirable (e.g. based on the COPERT database), and will definitely improve local model performance. However, introducing a detailed traffic model, including fleet composition, was considered outside the scope of this work.

At the very least, the author should discuss these potential issues and lay out future steps to resolve these problems. In the best case scenario, it would be good to see some sensitivity studies testing the modelled NO2 sensitivity to inclusion of these different classes.

New validation results in the Supplementary Material show that the current method of interpolated traffic flow predicts the traffic volume within a 50% error margin at most locations. Better results are obtained when more counting locations are available, or when they are selected strategically around crossings and access roads. It is now remarked in the Discussion & Conclusion section that improved traffic emission models should take local differences in local fleet composition into account.

Thirdly, I feel that the manuscript could benefit from some more detail on how the error characteristics of the observations were derived. Estimating uncertainties from reference instruments and low-cost sensors on its own is a difficult subject and the paper does not provide the reader with information on how these were estimated or how such uncertainties were then transformed into error characteristics suitable for ingestion in the OI scheme. Such a discussion should be included in the paper and I believe this would strengthen the authors conclusions.

The section on observations has been elaborated hereupon. The accuracy of reference instrumentation is determined following the EN 14211 standard (now referenced in the section on observations), which includes all aspects of the measurements method: uncertainties in calibration gas and zero gas, interfering gases, repeatability of the measurement, derivation of $NO_2$ from $NO_X$ and NO, and averaging effects. The error in the low-cost sensors is determined by side-by-side comparison against reference instruments. Details can be found in the referenced Mijling et al. (2018) paper.

Finally, given that 90% of the paper deals with modelling and data assimilation, I do find the choice of AMT for this manuscript slightly puzzling and I feel that the paper would probably be better suited for a journal more focused on modelling or general air quality issues. However since the editor has accepted the paper to AMTD I assume that the material is considered suitable for the journal.

AMT was chosen because the study describes an observation-driven framework which can be used for processing air quality measurements of different sources. Therefore, it was felt that it is of interest to the air quality measurement community. As the involved data assimilation sits on top of both observation and model, the current work would neither fully qualify for an air quality modelling journal. To put more emphasis on the importance of observations, Section 2.3 on observations is now put forward in the manuscript as an independent section.

**DETAILED COMMENTS**

L15: "Retina" - why is it called this? Include the full name if this is an acronym.
The algorithm's name reflects its ambition to produce high-resolution imagery; it is not an acronym. A simple name was preferred above a badly constructed or unpronounceable acronym.

L16/17: how are these percentages to be interpreted? Would be good to mention here how accuracy is defined. Something as simple as "... a typical accuracy (defined here as [...]) of 39%" or similar
Added "(defined here as the ratio between the root means square error and the mean of the observations)"

L23: "enhanced understanding of reference measurements". Please [missing]

L38: "adding value". I suggest you give an example of what you consider as adding value to the measurements or otherwise better write "exploiting the measurements"
Changed to "exploiting the measurements"

L39: In single-author papers it looks quite odd to use plural terms such as "our" and "we". Consider revising.
I replaced the inappropriate use of *pluralis majestatis* by the passive tense.

L43: I would add here that it depends on the mapping resolution and the pollutant. The required sampling density increases with the desired spatial resolution of the map. Furthermore, NO2 with its very sharp spatial gradients will always require a much denser network than for example mapping PM2.5 with its relatively smooth spatial gradients.
Added: "To obtain high-resolution information *of air pollutants with sharp concentration gradients*, (...)". From the first paragraph it is clear that is especially the case for NO2 concentrations, "which can vary considerably from street to street".

L60: The introduction/background section is missing a reference and a discussion of Tilloy et al (2013) (https://agupubs.onlinelibrary.wiley.com/doi/full/10.1002/jgrd.50233),who have essentially done the same as this paper (OI of point-based observations into an urban-scale AQ model).
The paper of Tilloy is now referenced twice.
In the Introduction: "*Tilloy et al. [2013] use the 3-hourly output of a well-developed implementation of the AMDS Urban dispersion model in Clermont-Ferrand, France, to assimilate in-situ NO2 measurements at 9 reference sites in an optimal interpolation scheme. With a leave-one-out validation they show a strong reduction in root mean square error of the time series after assimilation.*"

In Section 5.1 on the modelling of the error covariance matrix: "*Tilloy et al. [2013] choose to model the covariances depending on the road network. Error correlations are assumed to be high on the same road or on connected roads. For background locations, the correlation decreases fast in the vicinity of a road, while the error correlation between two background locations remains significant across a larger distance. The error covariances are kept constant in time, and taken independent of traffic conditions.*"

L61: Again, the name "Retina" comes a bit out of the blue. You should probably introduce here what the acronym stands for.
Retina is not an acronym, please see the answer above.

L70: I would be a bit careful with the term "calibration" in this context, given that it has a very specific meaning for measurements (both reference as well as sensors). Maybe reword or describe a bit more thoroughly what happens in this step.
I prefer to stick to *calibration* here, as I think that within the context of model calibration it is sufficiently clear that it refers to adjusting model parameters to best match the evaluation criteria.

L98: A reference to AERMOD would be useful here.
AERMOD has already been introduced and referenced shortly above (Cimorelli et al., 2004). This particular version of AERMOD has no specific scientific reference.

L99: "local equidistant coordinate system" - at this point you might as well give the actual projection you used. Presumably something UTM-like?
Clarified to: "*All coordinates are reprojected in a custom oblique stereographic projection (EPSG:9809) around the city center coordinate, such that the coordinate system can be considered equidistant at the urban scale.*"

L100: "road-following grid". This is used as if this a commonly known term, which in my opinion it is not. So first of all you might want to introduce this term a bit more carefully by saying something like "we use a road-following grid, which is essentially....". Secondly, to me it sounds a bit weird to use the term "grid" in this context, when you are basically talking about a spatially irregular and scattered set of receptor points with higher density along road links. I think the term grid should be reserved for a somewhat regular arrangement of cells.
The occurrences of *road-following grid* are replaced by *road-following mesh*.

L148-152: I realize that the goal of this paper is not to build the world's best model so a certain amount of simplification is expected, but interpolating traffic flow using IDW seems to be an incredibly crude method. How can this method possibly work?Between two loop counters there will likely be many road segments that either have much more or much less traffic than at the observation sites, so I fail to understand how simply interpolating here can lead to useful results. I think this section needs more detail on how this is carried out and a robust demonstration that the chosen methods are meaningful.
This concern is shared with Reviewer #1. In the revised article, the traffic model is assessed in two different approaches, presented in the Supplementary Material, and mentioned at the end of Section 3.2.1: a leave-one-out validation to study the error in local traffic flow estimations, and a concentration validation study of dispersion simulations done under different traffic scenarios. The results show that for this counting network IDW predicts the

traffic volume within a 50% error margin at most locations. The model simulations show that using inferior traffic data is partly compensated by the calibration dynamics, at the expense of less pronounced concentration gradients.

L164: Don't they assimilate UTD data? In that case it wouldn't be a day old but just a few hours (maybe better write "up to a day old" or so).
Clarified to: "*The analysis of the ensemble is based on the assimilation of up-to-date (UTD) air quality observations provided by the European Environment Agency (EEA).*"

L230: This should be Figure 3b? Also, I think this Figure should be discussed a bit more (maybe in the discussion section?) for example with respect to potential reasons for the difference between model and observations, particularly for the highway location.
Reference to Figure 3b now included. Differences between model and observations are now mentioned in the discussion section and mainly attributed to its inability to resolve all small-scale structures provoked by local built-up area, and sketchy traffic emission modelling. The latter is especially true for highway location NL49007, which is very near this strong source.

L235: I recommend to remove the term "geostatistical" here. While OI is mathematically very similar to kriging-based techniques (and it can in fact be shown that it provides identical results to kriging if the same inputs are used) it is not traditionally considered a part of the field of geostatistics. Geostatistics was developed in the mining and earth resources community (Matheron et al.), whereas OI was developed within the meteorological community (Gandin 1965).
Rephrased to "the interpolation technique of choice here is".

L241: Again, I would not use the term "grid" for what is essentially a set of irregular, scattered receptor points.
*Grid* replaced by *mesh*.

L265: I think it would be good to mention here that Statistical Interpolation/OI is essentially the same assimilation scheme (just a different mathematical framework) asprevious kriging-based approaches. The main advantage of OI over geostatistics (but also an added complexity) is that one has detailed manual control over the Pb covariance matrix, which allows for a more comprehensive specification of the area of influence for each contributing observation.
This valuable remark has been included in the beginning of the section, where OI is first introduced.

L287: extend -> extent (or maybe magnitude?); also reflect -> reflects
Corrected

L335-340: I think this section should be either left out entirely or expanded upon significantly. As it is currently it does not represent a robust demonstration that low-cost sensors add value to the system, since the effect has only been shown at a single site and not been analysed in detail. Demonstrating that the information from low-cost sensors can improve urban-scale air quality modelling is clearly a very worthwhile goal but this short section reads unfortunately more like an afterthought than a proper analysis.

This section has been expanded with new material: an additional analysis and discussion for 5 different assimilation scenarios. The different assimilation scenarios show that low-cost sensor data assimilation can improve the results locally, even in absence of reference data.

L354: I think it would also be worthwhile noting here that, while CAMS is definitely useful for providing background conditions and initial conditions, for NO2 the CAMS forecasts can be very misleading when interpreted at the local scale. The predicted diurnal cycle can often deviate substantially from that observed at urban AQ stations.
This valuable remark has been added in the discussion.

L356: Agreed. And in addition the higher resolution from a dispersion model is also much more appropriate than CAMS for applications such as exposure estimates etc.

L374: "Traffic data tend to be harder to obtain". That is very true (and maybe even an understatement) and is one of the most limiting factors in running local-scale dispersion models at random locations. Given that traffic is typically the most important source for NOx I have the suspicion that even the comparatively portable Retina methodology is likely to fail when no such traffic data is available at all. I think it would be worthwhile discussing here that at some point, if nearly all the crucial input data to Aermod is either of low quality or entirely missing, the resulting forecasted concentration fields will be so bad that any type of sophisticated data assimilation of observations is no longer very meaningful.
A traffic degradation study is now included in the Supplemental Material. From this can be seen that the calibration is partly capable of compensating for inferior traffic data by relating traffic emissions more to population density, at the expense of higher RMSE and lower correlation. In general, degraded input data and imperfections in the dispersion modelling will deteriorate the system's capability to resolve local structures; it will lower the effective spatial resolution of the simulations. In its extreme it will only describe the blurry urban background pollution contribution added to the rural background. Oppositely, with improved input data and atmospheric modelling, the effective resolution will improve. This insight has been added to the discussion.

L396: It might be more detailed, but is it really much better? This section is too qualitative to draw much of a conclusion. As I said above, I think the manuscript would benefit from a more robust analysis along these lines.
This is now better addressed in the section on the added value of low-cost sensors.

Figure 2: The caption should indicate more clearly that the thin lines represent the traffic at individual stations.
The caption is rewritten.

Figure 3a: These maps would benefit from some basic cartographic elements, e.g. a background map from OpenStreetmap similarly to Figure 7/8, scale bar, coordinates etc.
A scale bar and North arrow have been added. A cartographic background has not been added as the domain can now be inspected by comparison with new Figure 1.

Figure 5: "Units are in meters" - not all of them. I recommend to either label all axes properly or to have a more thorough caption describing the various elements of this busy Figure in

more detail. It would also be helpful to have labels for each subplot (a,b, c..) so that they can be better referred to in the caption.

The representation has been improved by removing distance units on the axes by scale bars, by better formulating plot titles, and adding grid lines. The figure caption has been reformulated.

Figure 6: I think it is a bit confusing that the time series is only for 8 days, whereas the scatter plots show an entire month of data. Why not show the time series also for the entire period? If it is a visualization issue, it could be plotted over multiple rows. Similar to my earlier comments I also think that the analysis here would benefit from looking at more than just a single station.

Figure 6 has been replaced by two validation examples, for a well performing location (NL49012) and a worse performing location (NL49014). The time series plots have been removed, regarded as redundant as the performance can also be read from the scatter plots. Bar plots with error distributions have been added to better illustrate the effect of the assimilation. The validation plots for all reference locations are included in the Supplementary Material.

Figure 8: "IJ-tunnel" Should be marked on the map since non-locals will not be familiar with this.

A new panel has been added showing the location of reference stations and low-cost sensors in the central area, and the location of the IJ-tunnel.